# CONFORMAL LANGUAGE MODELING

**Victor Quach**[1,*]    **Adam Fisch**[1,*]    **Tal Schuster**[2]

**Adam Yala**[3,4]    **Jae Ho Sohn**[4]    **Tommi Jaakkola**[1]    **Regina Barzilay**[1]

[1]CSAIL, MIT    [2]Google Research    [3]UC Berkeley    [4]UCSF

## ABSTRACT

In this paper, we propose a novel approach to conformal prediction for language models (LMs) in which we produce prediction sets with performance guarantees. LM responses are typically sampled from a predicted distribution over the large, combinatorial output space of language. Translating this to conformal prediction, we calibrate a *stopping rule* for sampling LM outputs that get added to a growing set of candidates until we are confident that the set covers at least one acceptable response. Since some samples may be low-quality, we also simultaneously calibrate a *rejection rule* for removing candidates from the output set to reduce noise. Similar to conformal prediction, we can prove that the final output set obeys certain desirable distribution-free guarantees. Within these sets of candidate responses, we also show that we can also identify subsets of individual components—such as phrases or sentences—that are each independently correct (e.g., that are not "hallucinations"), again with guarantees. Our method can be applied to any LM API that supports sampling. Furthermore, we empirically demonstrate that we can achieve many desired coverage levels within a limited number of total samples when applying our method to multiple tasks in open-domain question answering, text summarization, and radiology report generation using different LM variants.

## 1    INTRODUCTION

Language models (LMs) have emerged as powerful tools for solving natural language processing (NLP) tasks. Given an input prompt, LMs generate a response from some predicted distribution over output text sequences. For modern models, these generations are often coherent and contextually relevant. At the same time, these generations can still contain mistakes, and lack certain aspects of robustness and reliability in terms of providing accurate, trustworthy predictions (Jones and Steinhardt, 2022; Krishna et al., 2021; Lin et al., 2022a; Mallen et al., 2022; Srivastava et al., 2022; Wang et al., 2023). Unfortunately, quantifying the uncertainty in LM outputs has remained a major challenge.

Conformal prediction is a popular model-agnostic and distribution-free method for creating prediction sets that contain the correct answers with high probability (Angelopoulos et al., 2023; 2021a;b; Bates et al., 2020; Lei et al., 2018; Romano et al., 2019; Vovk et al., 2005). Applying conformal prediction to generative models such as LMs, however, is challenging due to (a) the unbounded nature of their output space (i.e., all possible text sequences), and (b) the limited available (tractable) mechanisms for exploring all possible predictions. In particular, LMs can typically only approximately search or sample candidate responses. Furthermore, while several possible responses might be acceptable (e.g., correct or factual), small differences can result in abrupt changes in coherence or meaning.

In this paper, we propose an extension of conformal prediction that is tailored specifically to generative LMs. We only assume that the (potentially black-box) LM that is given to us can be used to sample diverse output sequences, together with their evaluated model likelihoods (i.e., the output token sequence logits). Like conformal prediction, our method offers a rigorous coverage guarantee by constructing prediction sets that, in our case, provably contain at least one acceptable response with high probability. Unlike conformal prediction, however, we do not enumerate the entire output space (which is impossible). Instead, we derive a calibrated *stopping rule* for sampling different outputs from the LM that get added to a growing output set of candidates, until we are confident that the

---

*Equal contribution. AF is now at Google DeepMind. Correspondence to `adamfisch15@gmail.com`.

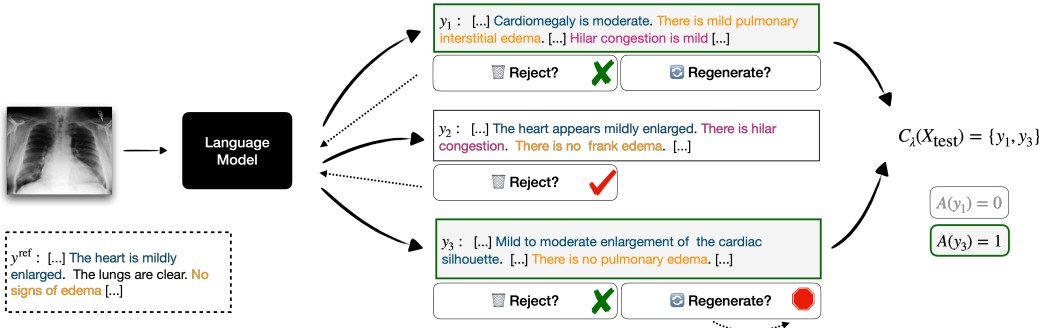

Figure 1: Our conformal procedure samples candidate outputs from some blackbox LM until a *stopping rule* is reached. Each sample is added to the output conformal set if it meets both a minimum estimated quality and a diversity criterion. The procedure is calibrated to stop when at least one candidate $y$ from the conformal set is admissible ($A(y) = 1$) with high probability. In this example, samples $y_1$ and $y_2$ are in-admissible because they hallucinate the presence of "edema" (in orange) and "hilar congestion" (in magenta), respectively. The minimal output set includes $y_3$, which is admissible.

output set is sufficient. Since not all samples from the LM may be high quality (e.g., some may be redundant, incoherent, or have lower confidence), we also simultaneously calibrate a *rejection rule* for removing candidates from the output set—while still ensuring that our coverage bound is not violated. This gives the benefit of making our output sets not only accurate, but also precise (i.e., small).

To more concretely describe the exact type of guarantee that we provide, suppose we have been given a calibration set $\mathcal{D}_{\mathrm{cal}} = (X_i, A_i) \in \mathcal{X} \times \mathcal{A}$, $i = 1, \ldots, n$ of independent and identically distributed (i.i.d.) prompts and "admission" functions (see also (Fisch et al., 2021a)). Here, $A_i$ is a binary random function that measures whether or not a generation $y \in \mathcal{Y}$ for prompt $X_i$ is "good enough" (i.e., $A_i(y) = 1$). Note that randomness in $A_i$ can come from implicit random covariates—such as relying on a random annotated reference, $Y_i^{\mathrm{ref}}$, to compare the candidate $y$ to. Figure 1 illustrates a setting where $X_i$ is an X-ray to automatically analyze and produce a report for, while $A_i$ extracts individual findings from each generated report and checks if they correspond to those given by an expert radiologist. Let $X_{\mathrm{test}}$ be a new i.i.d. test prompt. Using $\mathcal{D}_{\mathrm{cal}}$ to guide our choice of hyper-parameters $\lambda \in \Lambda$, for any $\epsilon, \delta \in (0, 1)$, our goal is to generate a set of samples $\mathcal{C}_\lambda(X_{\mathrm{test}}) \subseteq 2^{\mathcal{Y}}$ that satisfies

$$\mathbb{P}\Big(\mathbb{P}\big(\exists y \in \mathcal{C}_\lambda(X_{\mathrm{test}}) \colon A_{\mathrm{test}}(y) = 1 \mid \mathcal{D}_{\mathrm{cal}}\big) \geq 1 - \epsilon\Big) \geq 1 - \delta. \tag{1}$$

The outer and inner probabilities are over the draws of $\mathcal{D}_{\mathrm{cal}}$ and $(X_{\mathrm{test}}, A_{\mathrm{test}})$, respectively. $\epsilon$ is our error tolerance, while $\delta$ controls for the sensitivity of our algorithm with respect to calibration data.

While Eq. (1) stipulates the existence of at least one "acceptable" generation in $\mathcal{C}_\lambda(X_{\mathrm{test}})$, it does not tell us much about individual responses, $y \in \mathcal{C}_\lambda(X_{\mathrm{test}})$. Additionally, longer generations are often composed of multiple statements. In our radiology setting, a report may contain multiple findings, such as *"Cardiomegaly is moderate. There is mild pulmonary interstitial edema."* We futher identify a subset of confident components that would independently be categorized as being correct (given another admission function $A_{\mathrm{test}}^c$, this time operating over generation fragments). For example, we might predict that *"Cardiomegaly is moderate."* is correct, but perhaps not *"There is mild pulmonary interstitial edema."* This can not only be useful in catching incorrect statements, but can also help identify independently correct parts of a larger generation, even when the overall quality of the full generation is poor. Like Eq. (1), we calibrate this process such that it gives accurate results with high probability.

**Contributions.** In summary, our main results are as follows:

- We bridge the gap between conformal prediction and LMs by calibrating the *sampling* of output sets, rather than enumerating and selecting candidate responses directly from the output space;

- We extend multi-label conformal prediction to identify confident components of long generations;

- Though limitations apply, we demonstrate valid risk control on multiple diverse tasks with different LMs, while still retaining meaningful output sets that are efficient and precise compared to baselines.

## 2 RELATED WORK

**Conformal prediction and risk control.** Our work adds to the rich collection of tools for uncertainty estimation and risk control for machine learning algorithms (Angelopoulos et al., 2023; 2021a; Barber et al., 2021; Bates et al., 2020; Fisch et al., 2022; Gupta et al., 2020; Lei et al., 2013; 2018; Vovk, 2002; Vovk et al., 2015; 2017, *inter alia*). These techniques were previously extended and applied in the language domain to classification with finitely-many classes (Fisch et al., 2021a;b; Jones and Steinhardt, 2022), to token-level predictions (Dey et al., 2022; Ravfogel et al., 2023), and to reliably accelerate LMs (Laufer-Goldshtein et al., 2023; Schuster et al., 2021b; 2022b). Here, we address the emerging challenge of providing reliable prediction sets for unbounded, free-text generation—which previous methods are unequipped for. The distribution-free, finite-sample performance guarantees that we derive are similar to those given by prediction sets or regression intervals in standard conformal prediction (Angelopoulos et al., 2023; Papadopoulos et al., 2002; Vovk et al., 2005), but with slightly relaxed "correctness" criterions (Cauchois et al., 2022; Fisch et al., 2021a). In particular, we build on the groundwork set by Angelopoulos et al. (2021a), which provides a general methodology for calibrating any risk function that is controllable via some low-dimensional hyper-parameter configuration. We extend their framework to handle sampling-based algorithms that can effectively be used for LMs, and that, critically, do not require enumerating the full output space (which is intractable in our case). Most relevant to our work in LMs, other recent approaches have built on conformal principles to construct confidence intervals for generative diffusion models over images (Horwitz and Hoshen, 2022; Teneggi et al., 2023). These methods do not directly translate to LMs, however, as they only provide non-combinatorial confidence intervals at the pixel-level.

**Uncertainty estimation in LMs.** As the use of LMs in-the-wild quickly grows, there is increasing interest in obtaining and expressing meaningful confidence estimates for each output. Recent studies show that the logits of out-of-the-box LMs tend to exhibit overconfidence, even when wrong (Desai and Durrett, 2020; Kadavath et al., 2022; Miao et al., 2021; Vasconcelos et al., 2023). Recent alignment techniques degrade this even further (Kadavath et al., 2022; OpenAI, 2023). Most current mitigation approaches focus on introducing linguistic cues (Lin et al., 2022b; Zhou et al., 2023) or post-hoc logit calibration (Jiang et al., 2021; Kadavath et al., 2022; Mielke et al., 2022; Zablotskaia et al., 2023). In this work, we develop similar techniques to improve the output of the underlying LM. Our methods are model agnostic and provide rigorous guarantees. Our conformal component selection (§4.4) also relates to recent self-consistency work that builds on the empirical observation that repeated similar samples are more likely to be correct (Mitchell et al., 2022; Wang et al., 2023), and cross-sample entailment can approximate uncertainty (Kuhn et al., 2023). Unlike previous work that uses a fixed number of re-samples and compares full outputs, we (a) introduce a dynamic stopping rule to reduce the number of samples, (b) extend this concept to semantically compare sub-components of long text outputs, and (c) conformalize the process to provide formal guarantees.

**Reliable generation.** It is common practice to post-hoc apply classifiers and filters on top of LM generations for various quality goals such as preventing toxicity (Gehman et al., 2020; Rauh et al., 2022; Welbl et al., 2021), verifying grounding against sources (Bohnet et al., 2023; Liu et al., 2023; Yue et al., 2023), or re-ranking the set of decoded outputs (Jiang et al., 2022). Our work provides a systematic and reliable approach for filtering or flagging poor-quality outputs—both at a full generation and component level—and can also readily incorporate additional signal from auxiliary classifiers. For example, we demonstrate in our experiments using off-the-shelf natural language inference (NLI) models (Bowman et al., 2015; Khot et al., 2018; Schuster et al., 2021a; Thorne et al., 2018; Williams et al., 2018; Zhang et al., 2019) to help guide the selection of individual, confident components in text summarization (i.e., sentences that are fully entailed by the larger text (Fabbri et al., 2022; Honovich et al., 2022; Laban et al., 2022; Schuster et al., 2022a)).

## 3 BACKGROUND

We begin with a review of conformal prediction and risk control (see also (Angelopoulos and Bates, 2022)). We use upper-case letters ($X$) to denote random variables; lower-case letters ($x$) to denote constants, and script letters ($\mathcal{X}$) to denote sets, unless otherwise specified. Proofs are in Appendix D.

Given a new example $x$, for every candidate label $y \in \mathcal{Y}$ standard conformal prediction either accepts or rejects the null hypothesis that the pairing $(x, y)$ is correct. The test statistic for this test is a *nonconformity measure*, $\mathcal{M}((x, y), \mathcal{D})$, where $\mathcal{D}$ is a dataset of labeled examples. Informally, a

lower value of $\mathcal{M}$ reflects that point $(x, y)$ "conforms" to $\mathcal{D}$, whereas a higher value of $\mathcal{M}$ reflects that $(x, y)$ does not. For example, a practical choice for $\mathcal{M}$ could be the model-based negative log likelihood, $-\log p_\theta(y|x)$, where $\theta$ are parameters fit to $\mathcal{D}$. Split conformal prediction (Papadopoulos, 2008) uses a separate training set $\mathcal{D}_{\text{train}}$ to learn a fixed $\mathcal{M}$ that is not modified during calibration or prediction. To construct a prediction set for the new test point $x$, the conformal classifier outputs all $y$ for which the null hypothesis (that pairing $(x, y)$ is correct) is not rejected. This is achieved by comparing the scores of the test candidate pairs to the scores computed over $n$ calibration examples.

**Theorem 3.1** (Split conformal prediction (Papadopoulos, 2008; Vovk et al., 2005)). *Let $(X_i, Y_i)$, $i = 1, \ldots, n+1$ be exchangeable random variables. Let random variable $V_i = \mathcal{M}(X_i, Y_i)$ be the nonconformity score of $(X_i, Y_i)$, where $\mathcal{M}$ is fixed. For $\epsilon \in (0, 1)$, define the prediction (based on the first $n$ examples) at $x \in \mathcal{X}$ as*

$$\mathcal{C}_\epsilon(x) := \big\{ y \in \mathcal{Y} \colon \mathcal{M}(x, y) \leq \text{Quantile}(1 - \epsilon; V_{1:n} \cup \{\infty\}) \big\} \tag{2}$$

*Then $\mathbb{P}(Y_{n+1} \in \mathcal{C}_\epsilon(X_{n+1})) \geq 1 - \epsilon$.*

Note that the coverage expressed in Theorem 3.1 is *marginal* over the draw of calibration and test data. The recent Learn Then Test (LTT) framework of Angelopoulos et al. (2021a) extends conformal prediction to control the expectation of any loss function (conditional on the draw of calibration data) by reframing hyper-parameter selection as a multiple hypothesis testing problem.

Specifically, let $L : \Lambda \to \mathbb{R}$ be any random function using a hyper-parameter configuration $\lambda$ in some space $\Lambda$. For example, we might have $L(\lambda) := \ell(X, Y; \lambda)$ for some fixed loss function $\ell$ with random inputs $(X, Y)$. Unlike conformal prediction, however, $\lambda$ can be multi-dimensional (e.g., consist of multiple thresholds). Let $L_i$, $i = 1, \ldots, n$ be an i.i.d. calibration set $\mathcal{D}_{\text{cal}}$ of random functions, and $L_{\text{test}}$ a random test function. Let $\epsilon \in \mathbb{R}$ be a tolerance for the test risk, $\mathbb{E}[L_{\text{test}}(\lambda)] \leq \epsilon$. LTT then identifies a random (depending on $\mathcal{D}_{\text{cal}}$) subset of parameters, $\Lambda_{\text{valid}} \subseteq \Lambda$, with the goal of guaranteeing

$$\mathbb{P}\left( \sup_{\lambda \in \Lambda_{\text{valid}}} \mathbb{E}[L_{\text{test}}(\lambda) \mid \mathcal{D}_{\text{cal}}] \leq \epsilon \right) \geq 1 - \delta, \tag{3}$$

where the outer probability is over the draw of $\mathcal{D}_{\text{cal}}$, and the inner expectation is over draws of $L_{\text{test}}$. This then implies that any $\lambda \in \Lambda_{\text{valid}}$ can be selected to control the risk of $L_{\text{test}}$. In short, this is achieved by associating the null hypothesis $\mathcal{H}_\lambda \colon \mathbb{E}[L_{\text{test}}(\lambda)] > \epsilon$ to each $\lambda \in \Lambda$. For each null hypothesis, we then use the calibration set to compute a super-uniform p-value $p_\lambda$ using concentration inequalities. Any multiple testing algorithm $\mathcal{T}(p_\lambda \colon \lambda \in \Lambda)$ that controls the family-wise error rate (FWER) can then be used to identify the subset of non-rejected $\lambda$, i.e., $\Lambda_{\text{valid}}$.[1] Note that it is possible that $\Lambda_{\text{valid}} = \varnothing$, in the case that we fail to identify any statistically valid solutions (and the desired risk may not even be achievable with any $\lambda$). In this situation, we set $\lambda = \texttt{null}$, and either reject the task, or provide a trivial solution (e.g., a classifier that provides all possible labels $\mathcal{Y}$).

**Theorem 3.2** (Learn Then Test (Angelopoulos et al., 2021a)). *Suppose $p_\lambda$ is super-uniform under $\mathcal{H}_\lambda$ for all $\lambda$. Let $\mathcal{T}$ be any FWER-controlling algorithm at level $\delta$. Then $\Lambda_{\text{valid}}$ satisfies Eq. (3).*

Defining $\mathcal{C}_\lambda(x) := \{y \in \mathcal{Y} \colon \mathcal{M}(x, y) \leq \lambda\}$, $\Lambda \subset \mathbb{R}$, and $L(\lambda) := \mathbf{1}\{Y \notin \mathcal{C}_\lambda(X)\}$ recovers a criterion similar to that of conformal prediction (though not marginal over $\mathcal{D}_{\text{cal}}$). Unfortunately, in either instantiation (LTT vs. conformal prediction) iterating over $y \in \mathcal{Y}$ is intractable for LMs, regardless of whatever calibration technique is ultimately used. Instead, in §4, we introduce our method for generating uncertainty sets by casting $\lambda$ as a configuration of a *sampling* algorithm, rather than a filter on the output space $\mathcal{Y}$. We then show that this randomized algorithm can still be calibrated with LTT.

## 4 CONFORMAL LANGUAGE MODELING

We now introduce our method for generating uncertainty sets for LMs. At a high level, our procedure consists of three main steps to sample and return an collection of plausible output predictions:

1. **Sample.** A new candidate response $y$ is sampled from our language model.

---

[1] A FWER-controlling algorithm at level $\delta$ is any procedure that accepts or rejects null hypotheses $\mathcal{H}_\lambda$, while ensuring that the probability of falsely rejecting any $\mathcal{H}_\lambda$, $\forall \lambda \in \Lambda$, is less than $\delta$.

---

**Algorithm 1** Conformal sampling with rejection

---

**Definitions:** $x$ is an input prompt, $\mathcal{F}$ is our set-based confidence function, $\mathcal{S}$ is our text similarity function, $\mathcal{Q}$ is our sample quality estimator, $\lambda$ is our threshold configuration, and $k_{\max}$ is our sampling budget. $p_\theta(y \mid x)$ is the conditional output distribution defined by our language model.

```
 1: function SAMPLE(x, F, S, Q, λ, k_max)
 2:     C_λ ← {}                                          ▷ Initialize an empty output set.
 3:     for k = 1, 2, . . . , k_max do
 4:         y_k ← y ∼ p_θ(y | x).                         ▷ Sample a new response.
 5:         if Q(x, y_k) < λ_2 then                       ▷ Reject if its estimated quality is too low.
 6:             continue
 7:         if max{S(y_k, y_j): y_j ∈ C_λ} > λ_1 then     ▷ Reject if it is too similar to other samples.
 8:             continue
 9:         C_λ = C_λ ∪ {y_k}.                            ▷ Add the new response to the output set.
10:         if F(C_λ) ≥ λ_3 then                          ▷ Check if we are confident enough to stop.
11:             break
12:     return C_λ
```

---

2. **Accept or reject.** The sample $y$ is added to the growing output set, as long as it is diverse (e.g., maximum overlap with any other element is $\leq \lambda_1$) and confident (e.g., the LM likelihood is $\geq \lambda_2$).

3. **Stop or repeat.** Using a set-based scoring function, we check if the confidence in the current set is $\geq \lambda_3$. If it is, then we stop and return the current set. Otherwise we return to Step 1.

$\lambda = (\lambda_1, \lambda_2, \lambda_3)$ is a configuration that we calibrate to find a valid setting, $\hat{\lambda} = (\hat{\lambda}_1, \hat{\lambda}_2, \hat{\lambda}_3)$, that controls the risk of our output sets. In the following, we more carefully define our setting and notation (§4.1), and then describe our sampling (§4.2) and calibration algorithms (§4.3). Then, in §4.4, we provide an additional extension for highlighting confident generation components—i.e., subsections of our full generations that are independently likely to be correct, even if the full generation is not.

## 4.1 FORMAL SETTING AND NOTATION

Let $\mathcal{V}$ be an alphabet (a non-empty, finite set of tokens such as {"a", "b", "c", . . .}) from which all possible output strings, $y$, are composed, i.e. $\mathcal{Y} := \bigcup_{n=0}^{\infty} \mathcal{V}^n$. We assume that we are given a generative model $p_\theta(y \mid x)$ that defines a conditional probability distribution given some input prompt $x \in \mathcal{X}$ which we can sample from to obtain output strings, $y \sim p_\theta(y \mid x)$. Following Fisch et al. (2021a), for every input prompt $x$, we also further assume access to some "admission" function $A: \mathcal{Y} \to \{0, 1\}$ that is used to measure the acceptability of a given sample $y$. Intuitively, $A$ tells us if an output is "good enough". We explore different tasks and admission functions in our experiments in §5. See Appendix A for an extended discussion of this setting and its assumptions. Given a sampled calibration set $\mathcal{D}_{\text{cal}}$, our goal is then to derive a configurable algorithm with input parameters $\lambda \in \Lambda$ for constructing a prediction $\mathcal{C}_\lambda$ that we can calibrate to satisfy Eq. (1). In the framework of LTT (refer to §3), this is equivalent to defining $L_i(\lambda) = \mathbf{1}\{\nexists y \in \mathcal{C}_\lambda(X_i): A_i(y) = 1\}$, and using $\mathcal{D}_{\text{cal}}$ to find a value $\hat{\lambda}$ such that $\mathbb{E}[L_{\text{test}}(\hat{\lambda})] \leq \epsilon$, with probability at least $1 - \delta$ over the draw of $\mathcal{D}_{\text{cal}}$.

## 4.2 CONFORMAL SAMPLING WITH REJECTION

Let $\mathcal{F}: 2^{\mathcal{Y}} \to \mathbb{R}$ be a set-based function that, for any set $\mathcal{C} \in 2^{\mathcal{Y}}$, gives a **set confidence score** for the event $\mathbf{1}\{\exists y \in \mathcal{C}: A(y) = 1\}$. $\mathcal{F}$ should not depend on $\mathcal{D}_{\text{cal}}$. Furthermore, let $\mathcal{S}: \mathcal{Y} \times \mathcal{Y} \to \mathbb{R}$ be a text-based similarity function (e.g., BLEU or ROUGE) that we use to **detect duplicates and preserve diversity** in $\mathcal{C}$, and $\mathcal{Q}: \mathcal{X} \times \mathcal{Y} \to \mathbb{R}$ an input-conditional text-based measure of an **individual response's quality**—such as the LM's likelihood function, $p_\theta(y \mid x)$. We define and test different instances of these functions in §5.2. We then adopt a sampling-based procedure that *grows* an output set, $\mathcal{C}_1 \subseteq \mathcal{C}_2 \subseteq \ldots \subseteq \mathcal{C}_{k-1}$, by repeatedly taking samples $y_k \sim p_\theta(y \mid x)$, and updating

$$\mathcal{C}_k := \begin{cases} \mathcal{C}_{k-1} \cup \{y_k\} & \text{if } \max\{\mathcal{S}(y_k, y_j): y_j \in \mathcal{C}_{k-1}\} \leq \lambda_1 \\ & \text{and } \mathcal{Q}(x, y_k) \geq \lambda_2, \\ \mathcal{C}_{k-1} & \text{otherwise.} \end{cases} \tag{4}$$

until the confidence after $k$ samples, $\mathcal{F}(\mathcal{C}_k)$, is $\geq \lambda_3$ (or some sampling budget $k_{\max}$ is reached).

As an intuitive, but toy, example, suppose we modeled $y_k \sim p_\theta(y \mid x), k = 1, 2, \ldots$ as a Bernoulli process, where each $y_k$ has the same probability of success $p$ that we assume (albeit unrealistically) that we know. For $X_{\text{test}}$, "success" is determined by the admission function, $A_{\text{test}}$. The confidence that our current set $\mathcal{C}_k$ contains at least one admissible answer (without rejection) then follows a geometric distribution, $\text{Geo}(p)$: all that remains is to compute the minimum number of samples to take such that Eq. (1) is satisfied. This is achieved by taking $\mathcal{F}(\mathcal{C}_k) = k$ and $\lambda_3 = \lceil \log(\epsilon)/\log(1-p) \rceil$.

Of course, in reality we do not know the probability of success $p$ for test examples. Furthermore, the samples $y_k$ are not independent, and since we are also able to observe their values, better strategies may exist to conditionally estimate $A(y_k) = 1$. Therefore, we allow $\mathcal{F}$ to be *any* set-based function—that we also pair with similarity function $\mathcal{S}$, and sample quality function $\mathcal{Q}$, for handling rejections. Pseudocode is given in Algorithm 1. We derive, calibrate, and test different variations of $\mathcal{F}$, $\mathcal{S}$, and $\mathcal{Q}$ in §5 and §6, respectively. Using $\lambda = (\lambda_1, \lambda_2, \lambda_3)$, we write $\mathcal{C}_\lambda(X_{\text{test}})$ to denote the final output set.

### 4.3 CALIBRATION WITH LEARN THEN TEST

Let $\Lambda$ be a finite set of configurations. For example, if searching for a value of $\lambda = (\lambda_1, \lambda_2, \lambda_3) \in [0, 1]^3$, we might consider the evenly-spaced set $\Lambda = \{\frac{i}{\kappa} : i = 1, \ldots, \kappa\}^3$ for some finite $\kappa \in \mathbb{N}$. For each $\lambda \in \Lambda$, LTT then requires computing a valid p-value $p_\lambda$, where $p_\lambda$ is a super-uniform random variable under $\mathcal{H}_\lambda$. Here, we can obtain valid p-values from the empirical risk on $\mathcal{D}_{\text{cal}}$,

$$\widehat{R}_n(\lambda) := \frac{1}{n} \sum_{i=1}^{n} L_i(\lambda), \quad \text{where } L_i(\lambda) = \mathbf{1}\{\nexists y \in \mathcal{C}_\lambda(X_i) : A_i(y) = 1\}, \tag{5}$$

**Lemma 4.1** (Binomial tail bound p-values). *Let $\widehat{R}_n(\lambda)$ be the empirical risk in Eq. (5), and let* $\text{Binom}(n, \epsilon)$ *denote a binomial random variable with sample size $n$ and success probability $\epsilon$. Then*

$$p_\lambda^{\text{BT}} := \mathbb{P}(\text{Binom}(n, \epsilon) \leq n\widehat{R}_n(\lambda)) \tag{6}$$

*is a valid p-value for $\mathcal{H}_\lambda : \mathbb{E}[L_{\text{test}}(\lambda)] > \epsilon$.*

When paired with any FWER-controlling algorithm $\mathcal{T}$ at level $\delta$, we obtain the set $\Lambda_{\text{valid}} \subseteq \Lambda$ by selecting all configurations for hypotheses $\mathcal{H}_\lambda$ that are rejected by $\mathcal{T}(p_\lambda^{\text{BT}} : \lambda \in \Lambda)$. If $\Lambda_{\text{valid}}$ is empty, then we abstain (i.e., return `null`). Otherwise, we are free to use any configuration in $\Lambda_{\text{valid}}$. We then select the one that empirically minimizes a weighted combination of the average final set size (after rejection) as well as the relative number of "excess" samples taken from our model (i.e., how many extra samples our algorithm takes *after* the first admissible answer has already been surfaced, proportional to the total number of samples). Specifically, let $S_\lambda(x)$ be the total number of samples taken, $S^*(x)$ be the oracle sample index $j$ of the first admissible generation (where $A(y_j) = 1$), and $\mathcal{C}_\lambda(x)$ be the final prediction set. Then, reusing $\mathcal{D}_{\text{cal}}$, we take

$$\hat{\lambda} = \operatorname*{argmin}_{\lambda \in \Lambda_{\text{valid}}} \frac{1}{n} \sum_{i=1}^{n} \left( \rho_1 |\mathcal{C}_\lambda(X_i)| + \rho_2 \frac{[S_\lambda(X_i) - S^*(X_i)]^+}{S_\lambda(X_i)} \right) \tag{7}$$

where $\rho_1, \rho_2 \in \mathbb{R}_{\geq 0}$ are hyper-parameters and $[\cdot]^+ \triangleq \max(\cdot, 0)$. We choose $\rho_1 = \rho_2 = 0.5$. As a consequence of LTT, the chosen $\hat{\lambda}$ (which is a random variable that depends on $\mathcal{D}_{\text{cal}}$) is risk-controlling.

**Theorem 4.2** (Sampling-based LTT). *Let $\hat{\lambda}$ be defined according to Eq. (7). Then the prediction $\mathcal{C}_{\hat{\lambda}}(X_{\text{test}})$ computed by Algorithm 1 satisfies Eq. (1).*

**Remark 4.3.** Given a finite $k_{\max}$, Algorithm 1 is guaranteed to terminate. Smaller $k_{\max}$ will, however, shrink the range of achievable $\epsilon$ (i.e., with $\hat{\lambda} \neq$ `null`). See Appendix C for additional discussion.

To efficiently search and test the higher dimensional $\lambda = (\lambda_1, \lambda_2, \lambda_3)$, we use the Pareto Testing procedure from Laufer-Goldshtein et al. (2023). Pareto Testing exploits structure in $\Lambda$ by first using a proportion of $\mathcal{D}_{\text{cal}}$ to find $\Lambda$'s Pareto-optimal frontier, and then iteratively validates promising configurations using Fixed Sequence Testing (Holm, 1979) on the remaining calibration data. See Appendix E.

---

**Algorithm 2** Conformal component selection

---

**Definitions:** $\mathcal{C}_\lambda$ is a prediction set, $\mathcal{E}$ is an algorithm for splitting candidates $y$ into components, $\mathcal{F}^c$ is a confidence estimator for individual components, $\gamma$ is our threshold configuration.

1: **function** SELECT($\mathcal{C}_\lambda, \mathcal{E}, \mathcal{F}^c, \gamma$)
2:     $\mathcal{C}_\gamma^{\text{inner}} \leftarrow \{\}$                                                      ▷ Initialize an empty output set.
3:     **for** $y \in \mathcal{C}_\lambda$ **do**                                          ▷ Iterate over full predictions.
4:         **for** $e \in \mathcal{E}(y)$ **do**                                ▷ Iterate over individual components.
5:             **if** $\mathcal{F}^c(e) \geq \gamma$ **then**
6:                 $\mathcal{C}_\gamma^{\text{inner}} \leftarrow \mathcal{C}_\gamma^{\text{inner}} \cup \{e\}$                 ▷ Keep only high-confidence components.
7:     **return** $\mathcal{C}_\gamma^{\text{inner}}$

---

## 4.4 CONFORMAL SELECTION OF INDIVIDUAL COMPONENTS

A caveat of language generation is that LM responses can be verbose, and composed of multiple components. We consider a "component" to be a logically defined subpart of a larger response, such as a series of phrases or propositions. For example, a radiology report like *"The heart is mildly enlarged. The lungs are clear."* can be broken down into two findings: *"The heart is mildly enlarged."* and *"The lungs are clear."* While Theorem 4.2 guarantees that complete, admissible generations do *exist* within our prediction sets, we cannot use it to make statements about the relative reliability of individual components within each response contained within that prediction set. Let $\mathcal{E}: \mathcal{Y} \to 2^{\mathcal{Y}}$ be a deterministic function that takes a text string and breaks it down into components. We implement $\mathcal{E}$ to be a simple sentence splitter. For every input $x$, we assume access to some *component-based* admission function $A^c: \mathcal{Y} \to \{0,1\}$ that is used to judge **individual components for correctness**. For example, $A^c$ may check if $e$ is entailed by another component $e' \in \mathcal{E}(y^{\text{ref}})$, where $y^{\text{ref}}$ is a human reference. Let $\mathcal{F}^c: \mathcal{Y} \to \mathbb{R}$ be a function that, for component $e \in \mathcal{Y}$, gives a **confidence score** for the event $A^c(e) = 1$. We then define the subset of components $\mathcal{C}_\gamma^{\text{inner}} \subseteq 2^{\mathcal{Y}}$ as

$$\mathcal{C}_\gamma^{\text{inner}}(x) := \Big\{ e \in \bigcup_{y \in \mathcal{C}_\lambda(x)} \mathcal{E}(y) \colon \mathcal{F}^c(e) \geq \gamma \Big\}. \tag{8}$$

Using $\mathcal{D}_{\text{cal}}$, we seek to calibrate $\gamma \in \Gamma$, such that for test pair $(X_{\text{test}}, A_{\text{test}}^c)$ and $\alpha, \delta \in (0,1)$,

$$\mathbb{P}\Big(\mathbb{P}\Big(A_{\text{test}}^c(e) = 1, \forall e \in \mathcal{C}_\gamma^{\text{inner}}(X_{\text{test}}) \mid \mathcal{D}_{\text{cal}}\Big) \geq 1 - \alpha\Big) \geq 1 - \delta. \tag{9}$$

The outer and inner probabilities are over the draws of $\mathcal{D}_{\text{cal}}$ and $(X_{\text{test}}, A_{\text{test}}^c)$, respectively. The new parameter $\alpha$ can be interpreted as the maximum rate of making *any* false positive predictions in which we select a component that is not in fact acceptable. Like $\mathcal{C}_\lambda$, we calibrate $\mathcal{C}_\gamma^{\text{inner}}$ using LTT, but seek to make $\mathcal{C}_\gamma^{\text{inner}}$ as *large* as possible, in order to maximize recall of correct components. Concretely, let $L_i^c(\gamma) = \mathbf{1}\{\exists e \in \mathcal{C}_\gamma^{\text{inner}} \colon A_i^c(e) = 0\}$ and let $\Gamma_{\text{valid}}$ be the set of non-rejected configurations found by LTT (again using binomial tail p-values). During calibration we define $\mathcal{C}_\gamma^{\text{inner}}(X_i)$ using an upper bound to $\mathcal{C}_\lambda(X_i)$, by simply taking the first $k_{\max}$ samples, $\{y_1, \ldots, y_{k_{\max}}\}$. This will allow us to conveniently decouple component calibration from set calibration. We then use the configuration that empirically maximizes the average number of confident components (where we again reuse $\mathcal{D}_{\text{cal}}$):

$$\hat{\gamma} = \underset{\gamma \in \Gamma_{\text{valid}}}{\operatorname{argmax}} \ \frac{1}{n} \sum_{i=1}^{n} |\mathcal{C}_\gamma(X_i)|. \tag{10}$$

**Proposition 4.4** (Component-based LTT). *Let $\hat{\gamma}$ be defined according to Eq. (10), where $\mathcal{C}_\gamma(X_i)$ uses $\mathcal{C}_\lambda(X_i) \equiv \{y_1, \ldots, y_{k_{\max}}\}$ during calibration. Then the prediction set of components, $\mathcal{C}_{\hat{\gamma}}(X_{\text{test}})$ computed by Algorithm 2 paired with any $\mathcal{C}_\lambda$ at test time with $\sup_x |\mathcal{C}_\lambda(x)| \leq k_{\max}$ satisfies Eq. (9).*

By the union bound, Eq. (1) and Eq. (9) hold simultaneously with probability $1 - 2\delta$.

## 5 EXPERIMENTAL SETUP

In this section, we briefly describe our experimental setup, including tasks, scoring functions, and metrics. We set $k_{\max} = 20$ for all experiments. See Appendix F for additional task and model details.

## 5.1 TASKS

**Radiology report generation.** As motivated in §1, we apply our method to chest X-ray radiology report generation using the **MIMIC-CXR** (Johnson et al., 2019) dataset. For our LM, we fine-tune an encoder-decoder architecture based on a pretrained ViT (Dosovitskiy et al., 2021) image encoder and a GPT2-small (Radford et al., 2019) text decoder. To judge admission, we use the popular Clinical Efficacy metric (Liu et al., 2019; Nicolson et al., 2022) to check if the 14 labels predicted by an auxiliary CheXbert (Smit et al., 2020) model on the generated report exactly match the labels predicted by the same CheXbert model for a reference report from a radiologist. Similarly, a component (here a sentence including a finding) is defined to be admissible if it has a `ROUGE-L` (Lin, 2004) score $\geq 0.4$ (picked empirically), when compared to any component directly extracted from the reference.

**News summarization.** We also apply our method to news article text summarization using the **CNN/DM** (Hermann et al., 2015) dataset. For our LM, we finetune a T5-XL (Raffel et al., 2020) model. We define a candidate generation to be admissible if it has a `ROUGE-L` score $\geq 0.35$, when compared to all available reference summaries from human annotators. Like MIMIX-CXR, we define a component to be admissible if it has a `ROUGE-L` score $\geq 0.4$ when compared to components extracted from human summaries. These thresholds are picked through manual validation.

**Open-domain question answering.** Finally, we apply our method to open-domain QA using the **TriviaQA** (Joshi et al., 2017) dataset. Here we sample answers from LLaMA-13B (Touvron et al., 2023) in the few-shot setting $(k = 32)$, without any additional fine-tuning. Since answers are limited to one or few tokens, a candidate output generation is acceptable only if it exactly matches an annotated reference answer (after removing articles, casing, and punctuation). Furthermore, since the expected answers are short and fairly atomic, we do not evaluate component-level confidence for this dataset.

## 5.2 SCORING FUNCTIONS

Our method can support different quality functions $\mathcal{Q}$, similarity functions $\mathcal{S}$, and set scoring functions $\mathcal{F}$. We show that a straightforward approach is to simply use transformations on the model likelihoods (from token logits). We define $\mathcal{Q}(x, y) = p_\theta(y \mid x)$ using the likelihood function of the base LM, with length-normalization (Wu et al., 2016). We use `ROUGE-L` for $\mathcal{S}$. For $\mathcal{F}$, we experiment with:

- FIRST-K. As a baseline, we score a set by its size, $\mathcal{F}_{\text{FIRST-K}}(\mathcal{C}) = |\mathcal{C}|$, and do not use rejection. This corresponds to the number of samples taken, and follows the intuition from our toy example in §4.2.

- FIRST-K+REJECT. This variant uses our duplicate rejection component (§4.2), but like FIRST-K, scores a set by the total number of samples taken so far (where the number of samples is now $\geq |\mathcal{C}|$).

- MAX. The $\mathcal{F}_{\text{MAX}}$ scoring function stems from the intuition that a set is only as good as its best element, and defines $\mathcal{F}_{\text{MAX}}(\mathcal{C}) = \max\{\mathcal{Q}(y) \colon y \in \mathcal{C}\}$.

- SUM. Alternatively, we also use the sum of item-level scores: $\mathcal{F}_{\text{SUM}}(\mathcal{C}) = \sum_{y \in C} \mathcal{Q}(y)$.

## 5.3 METRICS

Our main motivation is to produce valid confidence sets that are also precise. To reflect this, we measure both the **loss** of our sets (which is guaranteed to satisfy our prescribed limits), as well as both (a) the **relative number of "excess" samples** taken from our model (including rejected samples, see also Eq. (7)), and (b) the ultimate **output size** of the prediction set (after rejection). Both metrics are important, as over-sampling wastes computation (or expensive API calls), while large output sets can be unwieldy to use and overall less helpful as an uncertainty quantification tool. We measure results and compute the normalized[2] AUC over the range of **achievable** $\epsilon$ or $\alpha$ (using a **fixed** $\delta = \mathbf{0.05}$), **excluding trivial** values (e.g., that a policy that always returns the first generation would satisfy).

## 6 EXPERIMENTAL RESULTS

We now present our main results. In all plots, solid lines give the mean over 100 trials and shaded regions show $+/-$ the standard deviation. Additional experimental results are reported in Appendix G.

---

[2]Specifically, we compute $\text{AUC}(f; a, b) = \frac{1}{b-a} \int_a^b f(\epsilon) d\epsilon$, where $[a, b]$ is the range of evaluated $\epsilon$ (or $\alpha$).

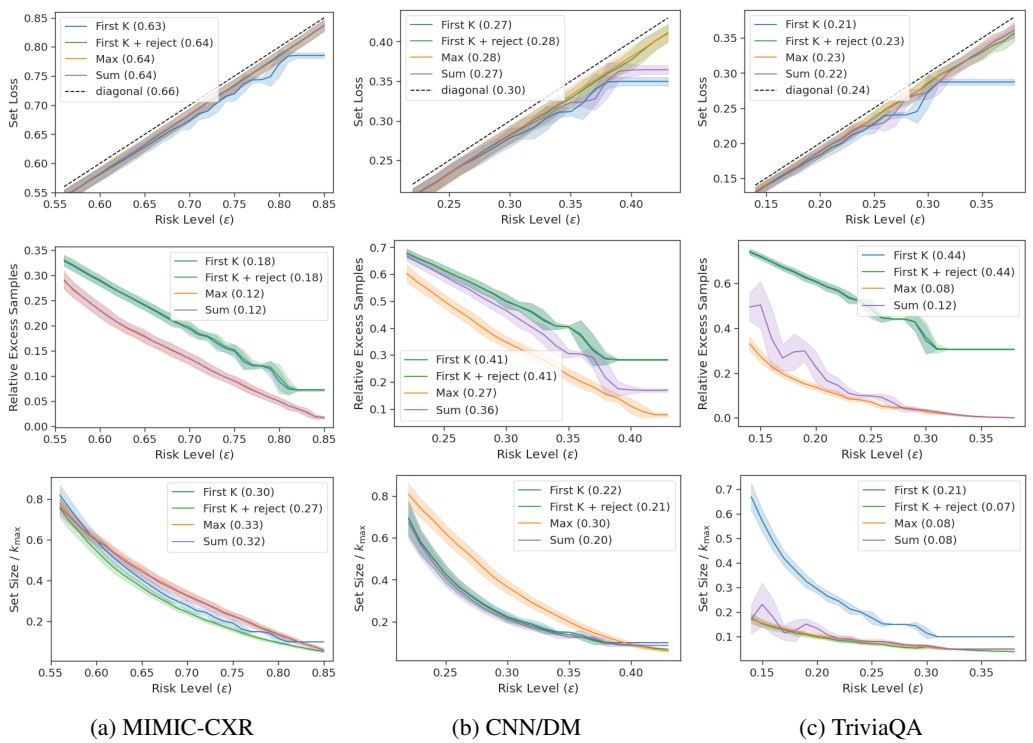

Figure 2: Conformal sampling results for $\mathcal{C}_\lambda$ a function of $\epsilon$. We report the loss, relative excess samples, and overall size (normalized by $k_{\max}$). We also report the AUC over achieved/non-trivial $\epsilon$.

**Validity of conformal sampling with rejection.** We demonstrate in Figure 2 that our conformal sampling approach matches our theory in practice, as the average set loss often matches but never exceeds the target risk level. Methods that have access to the model logits (MAX, SUM, FIRST-K-REJECT) are close to, but still below, the diagonal line, indicating that they are valid, but not conservative.

**Prediction efficiency.** The likelihood-based approaches outperform the uniform FIRST-K baseline across all three tasks. For example, as Figure 2c shows, the AUC of expected set size of MAX and SUM are both less than half the AUC of FIRST-K in the QA task. In tasks with longer output texts, FIRST-K produces competitive set sizes across all achievable $\epsilon$. However, on the relative number of excess samples metric, the MAX scoring function largely outperforms SUM and FIRST-K. FIRST-K+REJECT achieves similar size efficiency to the other rejection algorithms, but still lacks sampling efficiency.

**Individual components.** We evaluate two scoring functions $\mathcal{F}^c$ for conformal component selection. SPAN-LOGITS extracts the likelihood of a component using the base language model. However, as that likelihood is also conditioned on previous context, it may underestimate the score of a correct component that follows an incorrect component. We therefore also test an application-specific CLASSIFIER to assign a conformity score to each component. We compare to a RANDOM baseline which attributes a random score to any $(x, e)$ pair. Figure G.1 shows that by modeling components independently, we produce better (larger) sets. We include additional results in Appendix G and H.

## 7 CONCLUSION

Reliably using language models (LMs) in real-world tasks inevitably requires meaningful uncertainty quantification. In this paper, we introduced an approach to conformal prediction that allows a user to sample prediction sets from generative LMs with infinite, combinatorial output spaces, while retaining desirable statistical guarantees. Our method bridges the gap between standard conformal prediction and LM inference by calibrating a stopping rule for an algorithm that iteratively grows an output prediction set by sampling new generations (with rejection). Moreover, we can separately identify confident answer subcomponents. This can help users better understand the quality of long responses, which often include both correct and incorrect aspects. Empirically, we demonstrated that we can obtain efficient prediction sets, both in terms of size and total required samples.

## REPRODUCIBILITY STATEMENT

Code is available at https://github.com/Varal7/conformal-language-modeling. The codebase includes implementations of Algorithms 1 and 2, pre-processing code for our tasks, and functions for computing the metrics and producing all our plots and tables. In Section 5 and Appendix F we describe in detail all the datasets, language models, and scoring and admission functions we used, with references to downloading the public data and models, and all our hyper-parameters.

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

## A    ASSUMPTIONS

For clarity, we provide some additional discussion on our assumptions, and their implications.

**Assumption 1.** *Input prompts $x$ are i.i.d. for both calibration and testing.*

The inputs to our LM are considered to be randomly sampled from some fixed distribution. This is a reasonable assumption for many standard scenarios, such as the ones that we explore in our experiments, i.e.: questions for question answering, articles for summarization, and X-rays for radiology report generation. Importantly, however, this does **not** include multi-turn dialogue where successive prompts are dependent, or when there is distribution shift between calibration and testing. Additional modifications can be done to extend our calibration procedure to handle certain types of distribution shift (e.g., by defining new p-values that remain super-uniform under the target distribution using weighting), although we do not evaluate this direction in this work.

**Assumption 2.** *We can sample $y \sim p_\theta(y \mid x)$ using a language model API that accesses $p_\theta$.*

No other assumptions are placed on the LM itself or its sampling process. That said, two additional LM qualities also affect the performance of our method in practice:

Q1. There exists a good response that is expressible by the LM, i.e., $\exists y \in \mathcal{V}^*$ s.t. $A(y) = 1$. This simply is to say that all inputs are not impossible to answer appropriately.

Q2. The LM places high enough probability mass on good responses such that good responses are sampled within a tractable number of calls sufficiently often (i.e., $1 - \epsilon$ fraction of the time).

Without qualities Q1 and Q2, some settings of $k_{\max}$ and $\epsilon$ may be unachievable, and our algorithm will fail to return a risk-controlling configuration. Nevertheless, this **does not affect the validity of our algorithm**; it only affects its application. See Appendix C for a discussion on $k_{\max}$.

**Assumption 3.** *The admission function $A$ is a good proxy for assessing generation quality.*

Our guarantees are based on bounding the expected value of $A$ on future outputs. For this to be meaningful, $A(y) = 1$ should reflect that $y$ is a good sample. In our experiments, we manually design $A$ by using similarity metrics that compare possible responses to human references. For example, in our radiology report generation task, $x$ is the X-ray, $y$ is the report, and $p_\theta(y \mid x)$ is our image-to-text LM. Given $y$ and a "ground truth" report $y^*$ written by a radiologist (from the MIMIC-CXR dataset), we use the popular Clinical Efficacy metric (Liu et al., 2019; Nicolson et al., 2022) as a proxy for "admissibility", where we check if *all* of the 14 labels predicted by an auxiliary CheXbert (Smit et al., 2020) model given $y$ exactly match the labels predicted by the same CheXbert model given $y^*$.

The admission function is flexible, however, and need not be automatic. For example, the most reliable admission function is to directly use *real users* to assess whether a generated sample is acceptable or not (or the majority vote of one or more human annotators, when given clear, consistent guidelines). Such a user-based calibration set would be ideal, but also often costly to obtain.

When automatic admission functions are needed, here we show that it is also sufficient to only require access to a *conservative* admission function, $\bar{A}\colon \mathcal{V}^* \to \{0, 1\}$, where $\forall y \in \mathcal{V}^*$ we have $\bar{A}(y) \leq A(y)$. For instance, $\bar{A}$ might measure exact match on a word-for-word basis between $y$ and $y^*$, instead of accounting for differences in dictation. We show that $\hat{\lambda}$ remains valid with respect to the "true" (but inaccessible) $A_{\text{test}}$ if conservative admission functions $\bar{A}_i$ were used during calibration.

**Corollary A.1** (Conservative sampling-based LTT). *Suppose that over $\mathcal{D}_{\text{cal}}$ we let $L_i(\lambda) = \mathbf{1}\{\nexists y \in \mathcal{C}_\lambda(X_i)\colon \bar{A}_i(y) = 1\}$ where $\bar{A}(y) \leq A(y)$, $\forall y \in \mathcal{V}^*$. Then $\mathcal{C}_{\hat{\lambda}}(X_{\text{test}})$ still satisfies Eq. (1).*

*Proof.* The following proof is analogous to that of Propostion 4.4. Let

$$\bar{L}(\lambda) = \mathbf{1}\{\nexists y \in \mathcal{C}_\lambda(x)\colon \bar{A} = 1\} \tag{11}$$

For all $y \in \mathcal{V}^*$, we have $\bar{A}_{\text{test}}(y) = 1 \implies A_{\text{test}}(y) = 1$, which implies that $\bar{L}(\lambda) \geq L(\lambda)$ for all $\lambda$. This implies that for any choice of $\mathcal{D}_{\text{cal}}$

$$\mathbb{E}[\bar{L}_{\text{test}}(\hat{\lambda}) \mid \mathcal{D}_{\text{cal}}] \geq \mathbb{E}[L_{\text{test}}(\hat{\lambda}) \mid \mathcal{D}_{\text{cal}}]. \tag{12}$$

Applying Theorem 4.2 gives that the left hand side is $\leq \epsilon$ w.p. $\geq 1 - \delta$. $\qquad\square$

## B    LIMITATIONS

Our work aims to provide rigorous, yet useful, uncertainty estimates for language models. This has important implications for the safety and reliability of deployed models that make decisions with real consequences. At the same time, definite limitations do exist for the algorithms presented here, in particular (a) the assumption of i.i.d. data, (b) an appropriate admission function $A$, and (c) having resulting $\mathcal{C}_\lambda$ that are not too large or expensive to obtain (e.g., requiring many samples). In the same vein, if $k_{\max}$, the maximum number of samples drawn, is too low, then many levels of $\epsilon$ will be unattainable, and the method will fail to find a valid configuration (it will return `null`). Finally, it is important to emphasize that the guarantees presented here are probabilistic in nature—and also do not necessarily hold when conditioned on a particular type of input. While setting $\delta$ and $\epsilon$ to low values is possible and decreases the changes of failures, it will also make the algorithm more conservative and potentially less useful. The admission function $A$ also requires careful construction. Nevertheless, these results can be improved by (a) plugging in better language models, (b) using higher signal confidence metrics (e.g., as opposed to raw logits), and (c) obtaining larger samples $\mathcal{D}_{\text{cal}}$ for calibration.

## C    EFFECTS OF TRUNCATED SAMPLING ($k_{\max}$)

To be useful, it is critical to ensure that our sampling algorithm terminates in a reasonable number of steps. For this reason, we use $k_{\max}$ as a hard stop on the total number of samples we take from $p_\theta(y \mid x)$. Naturally, this also effects the achievable coverage that we can guarantee, as certain LMs

may require more than $k_{\max}$ samples to get a correct response for certain input examples. For each $k_{\max}$ there is therefore a *band* of achievable (non-trivial) $\epsilon$, that ranges from the error rate at first-1 to the error rate at first-$k_{\max}$ (where first-$k$ denotes the strategy of always taking the first $k$ samples for a fixed $k$). In our experiments, we set $k_{\max} = 20$, although the best practice is to empirically choose $k_{\max}$ using a development set along with an idea for how many samples one is willing to take in the worst case, which is primarily determined by the one's computational budget.

## D    PROOFS

### D.1    PROOF OF LEMMA 4.1

*Proof.* Let $X = \text{Binom}(n, \epsilon)$ and $Y = n\hat{R}_n(\lambda)$, which also has distribution $\text{Binom}(n, \epsilon')$, for some unknown success probability $\epsilon'$, as $L_i$'s are binary. We write $F_X(x)$ and $F_Y(y)$ to denote the CDFs of $X$ and $Y$, respectively. Under the null hypothesis $\mathcal{H}_\lambda : \mathbb{E}[L_{\text{test}}(\lambda)] > \epsilon$ (or equivalently, $\mathcal{H}_\lambda : \epsilon' > \epsilon$), $Y$ stochastically dominates $X$, i.e., $F_X(u) \geq F_Y(u), \forall u$. Let $Z = p_\lambda^{\text{BT}} = F_X(Y)$. Then

$$\mathbb{P}(Z \leq z) = \mathbb{P}(F_X(Y) \leq z) \tag{13}$$
$$\leq \mathbb{P}(F_Y(Y) \leq z) \tag{14}$$
$$= \mathbb{P}(Y \leq F_Y^{-1}(z)) \tag{15}$$
$$= F_Y F_Y^{-1}(z) \tag{16}$$
$$= z. \tag{17}$$

Therefore since $p_\lambda^{\text{BT}}$ is super-uniform, it is a valid p-value. $\quad\square$

### D.2    PROOF OF THEOREM 4.2

*Proof.* Since sampling is performed independently for each (i.i.d.) input prompt $X_i$ and admission function $A_i$, the set losses $L_i(\lambda)$ are also i.i.d. According to Lemma 4.1, $p_\lambda^{\text{BT}}$ is super-uniform under $\mathcal{H}_\lambda : \mathbb{E}[L_{\text{test}}(\lambda)] > \epsilon$. Given $\mathcal{T}$, a FWER-controlling algorithm at level $\delta$, we can apply Theorem 3.2 to identify $\Lambda_{\text{valid}}$ such that

$$\mathbb{P}\left( \sup_{\lambda \in \Lambda_{\text{valid}}} \mathbb{E}[L_{\text{test}}(\lambda) \mid \mathcal{D}_{\text{cal}}] \leq \epsilon \right) \geq 1 - \delta. \tag{18}$$

i.e.

$$\mathbb{P}\left( \inf_{\lambda \in \Lambda_{\text{valid}}} \mathbb{P}\left( \exists y \in \mathcal{C}_\lambda(X_{\text{test}}) \colon A_{\text{test}}(y) = 1 \mid \mathcal{D}_{\text{cal}} \right) \geq 1 - \epsilon \right) \geq 1 - \delta. \tag{19}$$

Therefore, Equation 1 holds for any $\lambda \in \Lambda_{\text{valid}}$. In particular, it holds for selecting $\hat{\lambda}$ by Eq. (7). $\quad\square$

### D.3    PROOF OF PROPOSITION 4.4

*Proof.* Let

$$\bar{L}^c(\gamma) = \mathbf{1}\left\{ \exists e \in \bigcup_{i=1}^{y_{k_{\max}}} y_i \colon A^c(e) = 0 \right\} \tag{20}$$

Since $\mathcal{C}_\lambda(x) \subseteq \{y_1, \ldots, y_{k_{\max}}\}$ for any $\lambda$ by definition, we have $\bar{L}^c(\gamma) \geq L^c(\gamma)$ for all $\gamma$. This implies that for any draw of $\mathcal{D}_{\text{cal}}$,

$$\mathbb{E}[\bar{L}_{\text{test}}^c(\hat{\gamma}) \mid \mathcal{D}_{\text{cal}}] \geq \mathbb{E}[L_{\text{test}}^c(\hat{\gamma}) \mid \mathcal{D}_{\text{cal}}]. \tag{21}$$

Similar to the proof of Theorem 4.2, since $\bar{L}^c(\gamma)$ is also binary, we can use Lemma 4.1 to show that $p_\gamma^{\text{BT}}$ is a valid p-value, and apply LTT to show that the left hand side is $\leq \alpha$ w.p. $\geq 1 - \delta$. $\quad\square$

## E   PARETO TESTING

We briefly review the Pareto Testing method for configuration selection, but refer the reader to Laufer-Goldshtein et al. (2023) for full details. Pareto Testing is a computationally and statistically efficient procedure that improves Fixed Sequence Testing (Holm, 1979), a common FWER-controlling procedure, by optimizing the ordering of configurations to test. The method consists of two stages:

**Stage 1: Constructing the Pareto frontier**

First we solve an unconstrained, multi-objective optimization problem in order to recover an approximate set of Pareto-optimal configurations, i.e., settings for which no other configuration exists that is uniformly better in all respects. Some of these objectives are meant to eventually be constrained (e.g., controlled to be $\leq \epsilon$), while others are meant to be optimized (e.g., find the smallest set).

Specifically, suppose that there are $c$ objectives we seek to control and $k$ objectives that we seek to optimize. In our setting, $c = 1$ (we would like to control coverage) and $k = 1$ (we would like to minimize a weighted combination of the number of samples and the set size per Eq. (7)). Let $\Lambda \triangleq \Lambda_1 \times ... \times \Lambda_m$ (here $m = 3$) be a multi-dimensional configuration space, and let $\mathbf{q}(\lambda) : \Lambda \to \mathbb{R}^{c+k}$ be a map from $\lambda$ to the values of the objective functions (both constrained and unconstrained), i.e.,

$$\mathbf{q}(\lambda) = \left[ \hat{Q}_1^{\text{opt}}(\lambda), \ldots, \hat{Q}_{c+k}^{\text{opt}}(\lambda) \right] \tag{22}$$

where $\hat{Q}_i^{\text{opt}}$ is the empirical average of the objective evaluated over a split of data, $\mathcal{D}_{\text{opt}}$ (e.g., the empirical coverage). Generally speaking, there is typically no single value of $\lambda$ that minimizes all objectives simultaneously. Instead, we find the Pareto frontier of all points that are not dominated (i.e., there exists another $\boldsymbol{\lambda}' \in \Lambda$ that is better in all respects):

$$\Lambda_{\text{par}} = \{\lambda \in \Lambda : \ \{\lambda' \in \Lambda : \ \lambda' \prec \lambda, \lambda' \neq \lambda \ \} = \emptyset\}. \tag{23}$$

$\Lambda_{\text{par}}$ then represents a set of configurations with optimal trade-offs (at least, according to the empirical values computed over $\mathcal{D}_{\text{opt}}$). Let $[\alpha_1, \ldots, \alpha_c]$ be a list of our target constraints for the first $c$ constrained objectives. We then sort $\Lambda_{\text{par}}$ by estimated p-values

$$p^{\text{opt}}(\lambda, \alpha) = \max_{1 \leq i \leq c} p(\hat{Q}_i^{\text{opt}}(\lambda); \alpha_i), \tag{24}$$

which we compute over $\mathcal{D}_{\text{opt}}$ (the same used to find the Pareto frontier, but separate from testing data). Different p-values can be used, see Angelopoulos et al. (2021a). Intuitively, this defines a sequence of configurations that are Pareto-optimal, ordered by the likelihood of them being able to satisfy all of our constraints. Since in this work we only have one constraint (coverage), this reduces to finding the sequence of Pareto-optimal configurations ordered from most to least likely to result in valid coverage.

**Stage 2: Fixed sequence testing**

The second stage is simple. Given the sequence of configurations identified in Stage 1, Stage 2 applies Fixed Sequence Testing. Concretely, given calibration data $\mathcal{D}_{\text{cal}}$, we sequentially test each configuration by checking if the maximum p-value for constrained objectives is greater than $\delta$, i.e.,

$$p^{\text{cal}}(\lambda, \alpha) = \max_{1 \leq i \leq c} p(\hat{Q}_i^{\text{cal}}(\lambda); \alpha_i) \geq \delta, \tag{25}$$

and stopping at the first configuration for which this inequality holds. The set of evaluated $\lambda$ for which Eq. (25) does not hold is then taken to be $\Lambda_{\text{valid}}$. It can be shown that this procedure is FWER-controlling at level $\delta$, and often powerful, as the constructed sequence of $\lambda$ generally allows for identifying a $\Lambda_{\text{valid}}$ with high-recall (i.e., we recover many of the valid configurations).

## F   ADDITIONAL EXPERIMENTAL DETAILS

In this section, we provide additional details regarding the experiments conducted for the three tasks discussed in Section 5. Our code will be released after the review process.

### F.1 Radiology report generation

**Dataset**  For the radiology report generation experiment, we utilized the labeled MIMIC-CXR and MIMIC-CXR-JPG datasets (Johnson et al., 2019). The MIMIC-CXR dataset can be accessed at https://physionet.org/content/mimic-cxr/2.0.0/ under the PhysioNet Credentialed Health Data License 1.5.0. Similarly, the MIMIC-CXR-JPG dataset is available at https://physionet.org/content/mimic-cxr-jpg/2.0.0/ under the same license.

We start with the standard splits prescribed in MIMIC-CXR-JPG. However, we further divide the training set into a train set and a dev set using a 0.9/0.1 ratio. The train set is used for training the model, using the validation set for early stopping. We then exclusively use the dev set for conformal prediction experiments. Subsequently, we filtered the dataset to include only anterior to posterior (AP) or posterior to anterior (PA) views and retained only one image per report. Furthermore, we removed examples where the report did not start with the phrase "FINAL REPORT" as these reports often contained a summary of the findings at the beginning, inadvertently leaking the answer we aimed to generate with the model. Table F.1 provides an overview of the resulting dataset.

| Split | Train | Dev | Validation | Test |
|---|---|---|---|---|
| Number of Images | 176,078 | 19,658 | 1,594 | 2,799 |
| Number of Studies | 176,078 | 19,658 | 1,594 | 2,799 |
| Number of Patients | 54,482 | 6,053 | 463 | 286 |

Table F.1: Dataset statistics for preprocessed MIMIC-CXR. The splits and preprocessing scripts are available within our code release. The train and validation split is used for to train the encoder-deocder model with early stopping. The dev set is used for conformal prediction. The test set is unused.

Each image was resized and cropped to a resolution of 224x224. Following prior methodology (Miura et al., 2021), we split each report into a *prompt* part and a *findings* part (which may also contain the *impressions* section) by identifying one of the following phrases: "FINDINGS AND IMPRESSION", "FINDINGS" or "IMPRESSION".

**Model**  The image encoder used in our experiment was a Vision Transformer (ViT) model pretrained on ImageNet-21k at a resolution of 224x224. Specifically, we utilized the `google/vit-base-patch16-224-in21k` model available in the Transformers library (Wolf et al., 2019). The text decoder was a GPT2-small model (`gpt2` on HuggingFace). We trained the model with a batch size of 128 distributed over 8 GPUs, resulting in a batch size of 16 per GPU. The AdamW optimizer was employed with $\beta_1 = 0.9$, $\beta_2 = 0.999$, and $\epsilon = 10^{-8}$. The learning rate was set to $5 \times 10^{-5}$. The training process consisted of 10 epochs, and the total training time on 8 RTX A6000 GPUs was approximately 11 hours.

**Generations**  Candidate reports were sampled from the model using default arguments from the Transformers library, i.e. top_k = 50, top_p = 1.0 and temperature = 1. Each generated report is then evaluated using a trained CheXbert model (Smit et al., 2020). The CheXbert model is available at https://stanfordmedicine.box.com/ under the Stanford Academic Software License. The CheXbert model labels each report for 14 conditions, assigning one of the following labels: "Blank," "Positive," "Negative," or "Uncertain."

To determine the admission of a candidate report, we compare it with a reference (human) report from the MIMIC dataset. If the candidate report matches all 14 labels of the reference report, the admission function returns 1; otherwise, it returns 0.

**Components**  We define a component as a sentence delimited by a period. The component-level admission function is defined based on how well a sentence "almost matches" one of the reference sentences. Two sentences are considered to "almost match" if their ROUGE score is above 0.4. If a sentence almost matches a reference sentence, the component-level admission function returns 1; otherwise, it returns 0.

```
Answer these questions

Q: Which American-born Sinclair won the Nobel Prize for Literature in 1930?
A: Sinclair Lewis
Q: Where in England was Dame Judi Dench born?
A: York
Q: In which decade did Billboard magazine first publish and American hit chart?
A: 30s
Q: From which country did Angola achieve independence in 1975?
A: Portugal
Q: Which city does David Soul come from?
A: Chicago
Q: Who won Super Bowl XX?
A: Chicago Bears
Q: Which was the first European country to abolish capital punishment?
A: Norway
Q: In which country did he widespread use of ISDN begin in 1988?
A: Japan
Q: What is Bruce Willis' real first name?
A: Walter
Q: Which William wrote the novel Lord Of The Flies?
A: Golding
Q: Which innovation for the car was developed by Prince Henry of Prussia in 1911?
A: Windshield wipers
Q: How is musician William Lee Conley better known?
A: Big Bill Broonzy
Q: How is Joan Molinsky better known?
A: Joan Rivers
...
```

Figure F.1: Truncated replication of the prompt used to generate answer on the TriviaQA dev set. The actual prompt contains 32 question-answer pairs.

### F.2    OPEN-DOMAIN QUESTION ANSWERING

We use the TriviaQA (Joshi et al., 2017) dataset available at https://nlp.cs.washington.edu/triviaqa/ under the Apache License Version 2.0. To generate candidate responses, we used LLaMA-13B (Touvron et al., 2023). We considered the closed-book setting, where the model does not have access to supporting text for answering the questions. We performed experiments in the few-shot setting by providing 32 example question-answer pairs sampled from the training set. A truncated prompt used for generating answers on the TriviaQA dev set is reproduced as an illustration in Figure F.1. Please note that the actual prompt used in the experiment contains 32 question-answer pairs.

For generating answers in the open-domain question answering task, we use the default Transformers parameters reported in the previous section. We extract an answer by considering the text until the first line break, comma, or period is encountered. We then normalize the answers: this involves converting the generated answers to lowercase, removing articles, punctuation, and duplicate whitespace. Generated answers are then compared using the exact match metric: an answer is considered correct only if it matches the provided answer exactly.

### F.3    NEWS SUMMARIZATION

We use the CNN/DM dataset (Hermann et al., 2015; See et al., 2017) that includes news articles from CNN and the Daily Mail paired with their human written summaries, and is available at https://github.com/abisee/cnn-dailymail under MIT License. We use the standard train set for finetuning, the validation set for selecting the best checkpoint, and the test set for all reported conformal experiments.

We use a T5 1.1 XL model, which includes roughly 3B parameters, and was further pretrained for 100k steps with a multilayer objective (Schuster et al., 2022b). We finetune the model on the train set for 200k steps with a batch size of 128 using 64 TPUv4 chips for approximately 40 hours. We use

the Adafactor (Shazeer and Stern, 2018) optimizer with a deacy rate of 0.8, initial learning rate of 0.001 and 1k warm-up steps.

To generate candidate responses, we use Nucleus sampling (Holtzman et al., 2020) with top-p set to 0.95, temperature 0.7, and maximum output length set to 256 tokens.

To get the response components we use a simple sentence spliter and treat each sentence as a component. As a classifier for evaluating the correctness of each component, we use an independent T5 XXL model trained on a mixture of NLI datasets (Honovich et al., 2022; Schuster et al., 2022a). Specifically, we leverage the model used in the TRUE benchmark (Honovich et al., 2022) and is available at https://huggingface.co/google/t5_xxl_true_nli_mixture. This model was trained on SNLI (Bowman et al., 2015), MNLI (Williams et al., 2018), FEVER (Thorne et al., 2018), SciTail (Khot et al., 2018), PAWS (Zhang et al., 2019), and VitaminC (Schuster et al., 2021a) to make a binary prediction of whether an hypothesis sentence is entailed by the given premise (in three-way datasets, the neutral class was merged with the negative class). We query the model with each component as the hypothesis, and the source summary as the premise, and measure the log-probability of predicting "entailment".

## F.4 DATASET DETAILS

| Dataset | Standard split | Size | Purpose |
|---------|---------------|------|---------|
| MIMIC | Train | 176,078 | Train the generative model |
| | Dev* | 19,658 | Calibration experiments |
| | Validation | 1,594 | Early stopping |
| | Test | 2,799 | Unused |
| TriviaQA | Train | 138,384 | Prompt LLaMA (32 samples) |
| | Validation | 18,669 | Calibration experiments |
| | Test | 17,210 | Unused |
| CNN/DM | Train | 287,113 | Train the generative model |
| | Validation | 13,368 | Early stopping |
| | Test | 11,490 | Calibration experiments |

Table F.2: We use the standard splits of each dataset and reserve unseen data for our calibration experiments. The only exception is for MIMIC where validation and test data are too small, so we reserve a subset of the official train set as unseen data for calibration. The asterisk marks that exception.

| Dataset | Split | Size |
|---------|-------|------|
| MIMIC | Calibration train | 2,000 |
| | Calibration val | 2,000 |
| | Calibration test | 15,658 |
| TriviaQA | Calibration train | 2,000 |
| | Calibration val | 2,000 |
| | Calibration test | 14,669 |
| CNNDM | Calibration train | 2,000 |
| | Calibration val | 2,000 |
| | Calibration test | 7,490 |

Table F.3: Additional splits for calibration experiments

For each dataset, we use the standard splits as shown in Figure F.2 and further split the data reserved for calibration experiments, as described in Figure F.3. The calibration "val" set was used in our earlier experiments to compare scoring functions. Some scoring functions required training (e.g. Platt scaling) and we used an additional calibration "train" set for that purpose. For final evaluation, we used the calibration "test" set to run 100 trials. For each trial, the calibration test data was split as follows: 10% is used to compute the Pareto frontier, 20% is used for Fixed Sequence Testing. The remaining 70% is used to measure test metrics (validity and efficacy).

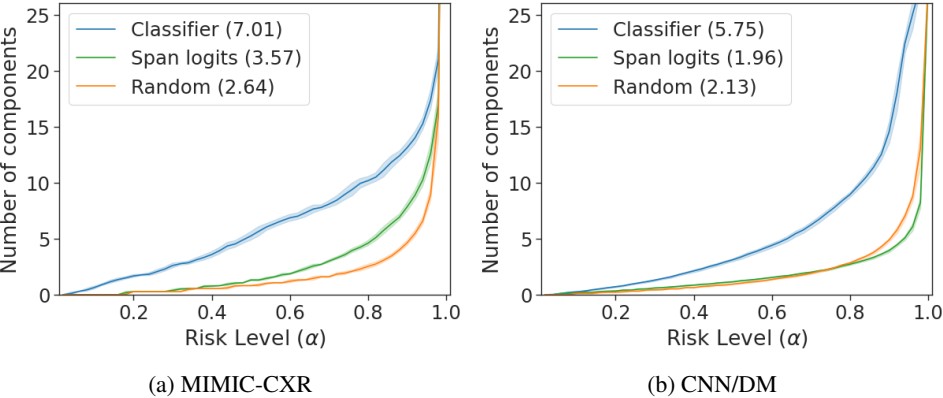

(a) MIMIC-CXR        (b) CNN/DM

Figure G.1: Conformal component selection results for $\mathcal{C}_\gamma^{\text{inner}}$ as a function of $\alpha$. We report the number of components in $\mathcal{C}_\gamma^{\text{inner}}$, which we want to maximize. We also report the AUC over $\alpha$.

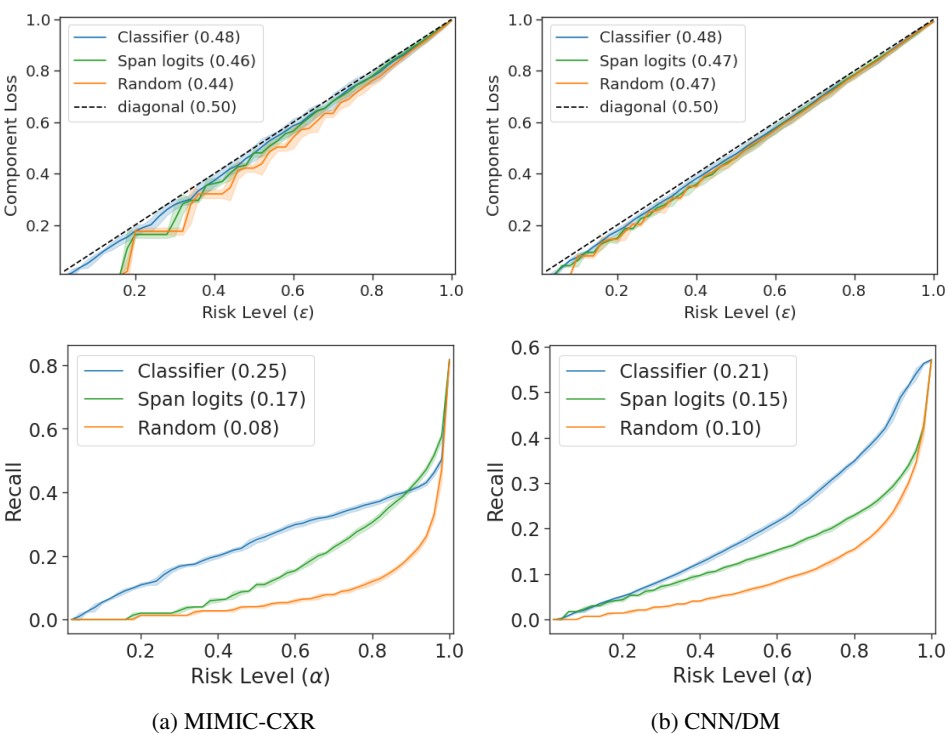

(a) MIMIC-CXR        (b) CNN/DM

Figure G.2: Component selection results for $\mathcal{C}_\gamma^{\text{inner}}$ as a function of $\alpha$. First row: validity curves. Second row: recall achieved by $\mathcal{C}_\gamma^{\text{inner}}$, which we want to maximize. We also report the AUC over $\alpha$.

## F.5   LENGTH-NORMALIZATION

For all tasks, we apply length-normalization (Wu et al., 2016) to the model logits, i.e. we compute:

$$\mathcal{Q}(x, y_k) = \exp\left(\frac{\log p_\theta(y_k|x)}{lp(y_k)}\right)$$

where

$$lp(y) = \frac{(5 + |y|)^{0.6}}{(5 + 1)^{0.6}}.$$

## G  ADDITIONAL RESULTS

We describe another metric useful to characterize the effectiveness of the components identified by our component selection method.

Given an input $x$ and a component set $\mathcal{C}\gamma^{\text{inner}}(x)$, we compute the recall by counting the number of reference sentences that "almost match" at least one element in $\mathcal{C}\gamma^{\text{inner}}(x)$. We then divide this count by the total number of reference sentences for that particular example. This gives us a measure of how much of the human reference is covered by the selected components. To obtain the expected recall, we average the recall values over all examples. The expected recall is reported in Figure G.2.

In particular, we observe that component sets generated using scoring functions based on an auxiliary CLASSIFIER outperform uncertainty measures based solely on the span logits provided by the model.

## H  QUALITATIVE RESULTS

We present qualitative results for radiology report generation and news summarization. In this section, we use the SUM method and consider $\mathcal{F}_{\text{SUM}}(\mathcal{C}) = \sum_{y \in C} \mathcal{Q}(y)$. The choice of $\alpha$ and $\epsilon$ is reported in Table H.7. We use 30% of the dev dataset (chosen uniformly at random) to determine $\hat{\lambda}$ as described in §4.3, and reserve the remaining 70% of the dataset for qualitative inspection. The corresponding values of $\hat{\lambda}$ and $\gamma$ are reported in Table H.7. Notably, the method produces $\lambda_2 = -\infty$ for the CNN/DM task, indicating that individual summaries are not rejected based on their quality but only for redundancy reasons.

In Figure H.1, an X-ray example is shown, depicting left basilar opacities while the rest of the X-ray appears normal. Table H.1 indicates that our method terminates the generation process after producing three samples. The third generation correctly identifies "apical scarring"; however, it mistakenly attributes it to the right lung instead of the left lung. This highlights a limitation of using CheXbert as the basis for the admission function, as its label granularity does not differentiate between left and right. Our component selection method accurately identifies several sentences that align with the reference report. These sentences are displayed in bold. Notably, our method avoids emphasizing low-confidence findings such as "right apical scarring" and instead focuses on the absence of an acute cardiopulmonary process.

A more challenging example is described in Figure H.2. The report mentions an enlarged heart, signs of cardiomegaly, and edema. Samples 4 and 5 correctly capture these findings but are considered incorrect due to the inclusion of "effusion." The conformal selection of components chooses not to highlight any sentences since none of them meet the confidence threshold defined by $\epsilon$.

In Tables H.3–H.6, we illustrate how our method continues sampling candidate summaries until the produced set is deemed acceptable. Specifically, Table H.3 demonstrates that the component selection process highlights the main idea while excluding minor ideas, which exist in multiple variations. Table H.4 exemplifies that the method stops after Sample 9, not because Sample 9 has the highest score, but because the sum of the scores collectively exceeds the target threshold of $\lambda_3 = 1.02$. Indeed, as shown in Table H.5, a higher individual score does not necessarily imply that a generation is more acceptable than one with a lower score. Finally, Table H.6 reveals a model failure, where the scores indicate high confidence in Sample 2, but the proposed generations are missing some main ideas from the reference summary.

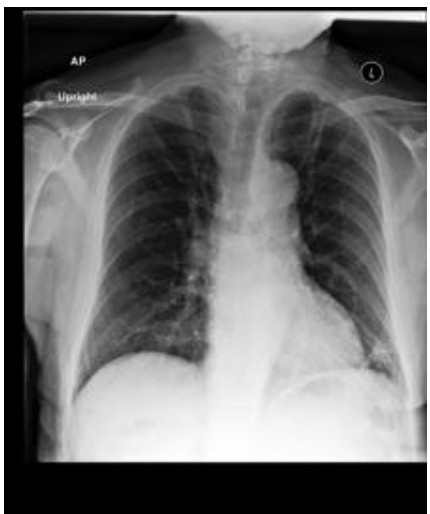

AP and lateral views of the chest. Streaky biapical and left basilar opacities are most compatible with scarring. The lungs are clear of confluent consolidation. There is no effusion. Cardiomediastinal silhouette is within normal limits. No acute osseous abnormalities detected. IMPRESSION: No definite acute cardiopulmonary process.

Figure H.1: Chest X-ray and reference radiology report for study id 55663120

Table H.1: Example prediction sets for study id 55663120, at $\alpha = 0.7$, $\epsilon = 0.7$. Bolded sentences are sentences that are chosen by the component selection process. Here we see that several phrases common to all generations, such as "AP and lateral views of the chest", are high confidence (and also correct). The finding about the cardiomediastinal silhouette is also confident. A completely correct generation, however, is not achieved until the third sample (after which the algorithm stopped).

| | Text | Score | Label |
|---|---|---|---|
| Sample 1 | **AP upright and lateral views of the chest provided.** There is chronic left lower lobe atelectasis. There is no focal consolidation concerning for pneumonia. **No large effusion or pneumothorax. The cardiomediastinal silhouette is normal.** Imaged osseous structures are notable for a mild compression deformity of an upper lumbar vertebral body. Degenerative change is seen in the right acromioclavicular joint. **IMPRESSION: No acute intrathoracic process.** | 1.45e-05 | False |
| Sample 2 | **AP and lateral views of the chest.** Diffusely calcified pleural plaques are identified bilaterally suggesting prior asbestos exposure. There is no pleural effusion. **The cardiomediastinal silhouette is within normal limits. No acute osseous abnormalities detected.** Surgical clips project over the upper abdomen. **IMPRESSION: No definite acute cardiopulmonary process.** | 1.98e-05 | False |
| Sample 3 | **AP and lateral views of the chest.** There is right apical scarring. Elsewhere, the lungs are clear. **The cardiomediastinal silhouette is within normal limits.** Atherosclerotic calcifications noted at the aortic knob. No acute osseous abnormality detected. **IMPRESSION: No acute cardiopulmonary process.** | 5.36e-03 | True |

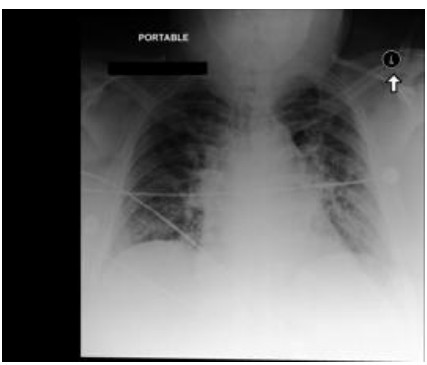

In comparison with the study of ___, there is continued enlargement of the cardiac silhouette with increasing fullness and indistinctness of central pulmonary vessels, consistent with worsening pulmonary edema. Mild asymmetry at the left base could represent developing aspiration or even infectious

Figure H.2: Chest X-ray and reference radiology report for study id 55770135

Table H.2: Example prediction sets for study id 55770135, at $\alpha = 0.7$, $\epsilon = 0.7$. No sentences are selected by the component selection process. As a symptom of low component confidence, most generations have low overlap in exact phrasing and findings. In this case, the sampling process *failed* to produce a correct response (though the probability of this happening overall is $\leq 0.3$).

| | Text | Score | Label |
|---|---|---|---|
| Sample 1 | In comparison with the study of ___, there is little overall change. Again there is enlargement of the cardiac silhouette with elevated pulmonary venous pressure and bilateral opacification is consistent with developing pulmonary edema or pneumonia in the appropriate clinical setting. The nasogastric tube again extends at least to the lower body of the stomach, where it crosses the lower margin of the image. | 1.96e-05 | False |
| Sample 2 | Compared to chest radiographs ___ through ___. Moderate pulmonary edema is exaggerated due to the low lung volumes, but is new, including mild interstitial edema and engorgement of the mediastinal veins. Mediastinal veins are still engorged, but not large. Pleural effusions are presumed, but not large. Indwelling right subclavian line ends in the low SVC. | 7.10e-08 | False |
| Sample 3 | Compared to chest radiographs ___ through ___. Moderate pulmonary edema is improving, although heart remains moderately enlarged and mediastinal veins are substantially dilated due to volume status. Bilateral pleural effusions are presumed, but not large. No pneumothorax. NOTIFICATION: I discussed the findings with the referring physician by telephone on ___ at 3:08 PM. | 2.42e-07 | False |
| Sample 4 | Compared to chest radiographs ___ through ___. Moderate to severe pulmonary edema has worsened. Moderate cardiomegaly is chronically large, exaggerated by lower lung volumes. Pleural effusions are small if any. No pneumothorax. | 2.35e-04 | False |
| Sample 5 | No previous images. The cardiac silhouette is enlarged and there is some indistinctness of pulmonary vessels consistent with mild elevation of pulmonary venous pressure. In view of the prominence of the pulmonary vasculature, it would be difficult to unequivocally exclude superimposed pneumonia, especially in the absence of a lateral view. | 4.69e-04 | False |

Table H.3: Example prediction sets for example from CNN/DM dataset, at $\alpha = 0.3$, $\epsilon = 0.7$. Bolded sentences are sentences that are selected by the component selection process. Missing sample indices represent samples that were rejected. Here we can observe that component selection highlights the main idea, while excluding minor details, which are lower confidence. A fully correct sample is not obtained until the 19th draw.

| | Text | Score | Label |
|---|---|---|---|
| Ref | Debris from boat to be dried, inspected and taken to landfill. The debris contained fish normally found in Japanese waters. The earthquake and tsunami hit Japan in March 2011. | | |
| Sample 1 | **Section of boat believed to be from 2011 Japan tsunami is found off Oregon coast .** Biologists say the environmental threat is small . | 3.62e-01 | False |
| Sample 2 | **Ship debris found off Oregon coast is suspected to be from 2011 Japan tsunami .** Biologists say the invasive species threat is small . | 2.63e-01 | False |
| Sample 3 | Ship fragment found off Oregon coast . It's suspected to be from 2011 Japan tsunami . **Yellowtail jack fish were found inside the boat .** | 1.71e-01 | False |
| Sample 7 | **Ship debris found off Oregon coast may be from 2011 Japan tsunami .** Yellowtail jack fish were found inside the vessel . | 2.76e-01 | False |
| Sample 12 | **Ship debris found off Oregon coast and towed to harbor .** Biologists say it poses no threat to the environment . | 1.22e-01 | False |
| Sample 13 | **Section of boat found off Oregon coast suspected to be from 2011 Japan tsunami .** Biologists say the boat fragment will be taken to a landfill . **Yellowtail jack fish, normally found in Japanese waters, will be taken to an aquarium .** | 2.63e-01 | False |
| Sample 16 | **Section of boat found off Oregon coast may be from 2011 Japan tsunami .** Biologists say the environmental threat is small . | 3.62e-01 | False |
| Sample 19 | **Section of boat found off Oregon coast suspected to be from 2011 Japan tsunami .** Biologists say the environmental threat posed by the boat is small . | 1.40e-01 | True |

Table H.4: Example prediction sets for example from CNN/DM dataset, at $\alpha = 0.3$, $\epsilon = 0.7$. Bolded sentences are sentences that are selected by the component selection process. Of note is how this example demonstrates that the method stops after Sample 9, not because Sample 9 has the highest score, but because the sum of the scores collectively exceeds the target threshold of $\lambda_3 = 1.02$.

| | Text | Score | Label |
|---|---|---|---|
| Ref | Jordan Ibe showed off the impressive dance move on his Instagram. The Liverpool star has broken into the first team during this campaign. Ibe is currently on the sidelines after suffering a knee injury. CLICK HERE for all the latest Liverpool news. | | |
| Sample 1 | Jordon Ibe has been a key player for Liverpool this season . The 19-year-old winger has been sidelined with a knee injury . Ibe has been performing the moonwalk while recovering from the injury . He posted a video on Instagram of him performing the move . **Liverpool face Newcastle in their next Premier League game .** | 2.63e-01 | True |
| Sample 2 | Jordon Ibe has been a key player for Liverpool this season . The 19-year-old winger has been sidelined with a knee injury . **Ibe took to Instagram to show off his skills performing the moonwalk .** | 1.60e-01 | False |
| Sample 3 | Jordon Ibe has been a key player for Liverpool this season . The 19-year-old is currently out with a knee injury . But Ibe showed off some impressive moves on his Instagram account . **The Liverpool winger posted a video of him performing the moonwalk .** | 2.63e-01 | True |
| Sample 5 | Jordon Ibe has been a key player for Liverpool this season . The 19-year-old took to Instagram to show off his dance moves . Ibe is currently out with a knee injury but is expected to return in the next few weeks . | 2.27e-01 | True |
| Sample 7 | Jordon Ibe has been a key player for Liverpool this season . The 19-year-old took to Instagram to show off his moves . **He posted a video of him performing the moonwalk .** Ibe is currently out with a knee injury but is expected to return soon . | 3.89e-01 | True |
| Sample 8 | Jordon Ibe has broken into the Liverpool first team this season . The 19-year-old took to Instagram to show off his skills performing the moonwalk . **Liverpool face Newcastle in their next Premier League game .** | 3.89e-01 | True |
| Sample 9 | **Jordon Ibe posted a video on Instagram of him performing the moonwalk .** The Liverpool winger is currently out with a knee injury . But Ibe is expected to return in the next few weeks . Liverpool face Newcastle in the Premier League on Sunday . | 2.92e-01 | False |

Table H.5: Example prediction sets for example from CNN/DM dataset, at $\alpha = 0.3$, $\epsilon = 0.7$. No sentences are selected by the component selection process. Here we can also see that a higher individual score does not necessarily imply that a generation is more acceptable than one with a lower score, which highlights why set calibration is critical. Note that this example also highlights a case in which a finer-grained component splitter might be useful: though no complete sentence is confident, certain topics, like "hormone therapy" appear to be.

| | Text | Score | Label |
|---|---|---|---|
| Ref | Craig Lister, 54, from Watford, is on hormone therapy for prostate cancer. This starves cancer cells by reducing levels of testosterone in the body. One side-effect is hot flushes: 'I have about ten to 12 flushes a day' | | |
| Sample 1 | Hormone therapy for prostate cancer starves cancer cells by reducing levels of testosterone in the body - and has the side-effect of hot flushes . It can't cure the cancer, but can help keep it under control . The majority of prostate cancers rely on the hormone to grow . Lowering the amount in the body means it's less able to spread . | 5.11e-01 | True |
| Sample 2 | Hormone therapy is usually given in the form of regular injections or implants . It can't cure the cancer, but can help keep it under control . The majority of prostate cancers rely on the hormone to grow . Lowering the amount in the body means it's less able to spread . | 9.10e-01 | False |

Table H.6: Example prediction sets for example from CNN/DM dataset, at $\alpha = 0.3$, $\epsilon = 0.7$. Bolded sentences are sentences that are selected by the component selection process. This example reveals a model failure, where the scores indicate high confidence in Sample 2, but the proposed generations are missing some main ideas from the reference summary.

| | Text | Score | Label |
|---|---|---|---|
| Ref | Independent Commission for Reform in Cycling (CIRC) concluded that the UCI colluded with Lance Armstrong to cover up allegations. Hein Verbruggen was criticised as events occurred under his leadership. Verbruggen has revealed he is having the report analysed by lawyers. | | |
| Sample 1 | **Hein Verbruggen is unhappy with the CIRC report verdict . The former head of international cycling has sent the report to lawyers .** Verbruggen claims the report is a 'character assassination' | 5.18e-01 | False |
| Sample 2 | **Hein Verbruggen is unhappy with the CIRC report verdict . The former head of international cycling has sent the report to lawyers . Verbruggen says he is having the report analysed by Swiss lawyers .** | 6.67e-01 | False |

Table H.7: Choice of $\epsilon, \alpha$ and corresponding $\lambda, \gamma$ for qualitative results presented in Appendix H.

| Dataset | MIMIC-CXR | CNN/DM |
|---|---|---|
| $\alpha$ | 0.7 | 0.3 |
| $\epsilon$ | 0.7 | 0.7 |
| $\lambda_1$ (similarity) | 7.37e-1 | 8.67e-1 |
| $\lambda_2$ (quality) | 2.47e-10 | $-\infty$ |
| $\lambda_3$ (set score) | 2.82e-4 | 1.02 |
| $\gamma$ (component threshold) | 2.04e-1 | 9.88e-1 |

