# OpenReview forum: "Conformal Language Modeling"
_ICLR.cc/2024/Conference — ICLR 2024 poster_

### Official Review · Reviewer_1uqq · 2023-10-26

**Soundness:** 3 good
**Presentation:** 3 good
**Contribution:** 3 good
**Rating:** 8
**Confidence:** 4

**Summary:**

This paper proposes an extension of the learn-then-test framework which could help generating responses from language models with risk control. It proposes a sampling pipeline which involves adding generations to a prediction set until the set-level confidence score is above a threshold. The whole pipeline involves using LTT to search for viable configs of three hyperparameters (for the quality of single generation, the rejection/diversity of single generation, and the quality of the prediction set). The experiments were carried out on three medical and benchmark datasets, suggesting that this pipeline generally is able to find feasible hyperparams.

**Strengths:**

This is a first (yet quite solid) step forward in conformal prediction in NLG. It proposes a general framework to sample responses from language models with some given criteria (admission function), and allows independent research to improve the various scoring functions (quality estimate, diversity rejection criteria, etc.) in the pipeline. Breaking down the full generation into different components is also a nice way to address the question of UQ for longer texts.

**Weaknesses:**

The main weakness is the use of ROUGE-L as a proxy of the quality of the generation. This only captures the the lexical similarity between the generation and the reference, which undermines the experiments quite a bit, in the following sense:
1. the LTT framework might return null hyperparameters, which might happen more often should the admissible function by the human be stricter than ROUGE
2. ROUGE was also used in the pipeline (detecting duplicate) so it's kind of "being its own judge"

Another potential improvement is to include a stronger baseline than first-K - for example, the duplicated responses (maybe even basing on rouge) should at least be deleted?

**Questions:**

1. How are the thresholds 0.4 and 0.35 for ROUGE-L determined? What does "These thresholds are picked through manual validation" mean exactly?

---

> ### Author Response · Authors · 2023-11-21
>
> We thank the reviewer for their careful review and thoughtful comments. We provide answers to specific questions and remarks (quoted) below.
>
> > The LTT framework might return null hyper-parameters, which might happen more often should the admissible function by the human be stricter than ROUGE
>
> This is true, however, the opposite is also true when the human evaluation is less strict than ROUGE at a high threshold (since by the same argument, the fact that ROUGE is sensitive to exact wording agreement between the generation and the reference can also lead to it being conservative).
>
> > ROUGE was also used in the pipeline (detecting duplicate) so it's kind of "being its own judge"
>
> Yes, ROUGE is also used in the pipeline, but this **does not affect the validity of the calibration process** (calibration is still computed based on similarity to the labeled answers only). In fact, we argue that having a similarity function that is directly connected to the final evaluation via transitive-like properties is a good thing. That is, if "$=$" is defined to be our similarity operator,  $A$ and $B$ are two generations where $A = B$, and $C$ is the target generation, we can choose to reject either $A$ or $B$ without incurring any increased risk. Of course, ROUGE-L and other proxies don't necessarily satisfy this, which is why we must calibrate thresholds simultaneously.
>
> > What does "These thresholds are picked through manual validation" mean exactly?
>
> The ROUGE-L thresholds for the admission functions were selected by the authors by manually inspecting a sample of random examples (~O(100)), evaluating ROUGE-L, and selecting the minimum value of ROUGE-L such that all samples with ROUGE-L greater than this threshold were acceptable. "Acceptable" in this context means that the generations were faithful and complete with respect to the perceived ground truth (which is inferred from the input and the collection of annotations available).
>
> Of course, this inevitably includes some error, as described in the paper. Though outside the scope of this paper, uncertainty in these labels can also be incorporated though methods like those explored in Angelopoulos et. al., 2023 and Stutz et. al., 2023. What makes a generation "acceptable" can also be better codified—for example, by following methods like Nenkova and Passonneau, 2004.
>
> In general, as described in our limitations and assumptions section, figuring out what makes a generation "acceptable" requires some careful thought and domain expertise. Still, fortunately our framework is general—and we can still bound the risk for any instantiation of $A(y)$. The design we used in this paper is illustrative, in addition to being reasonably effective in of itself.
>
> > Another potential improvement is to include a stronger baseline than first-K - for example, the duplicated responses (maybe even basing on rouge) should at least be deleted?
>
> Thanks for the suggestion—we added a variant with rejection. Please refer to our [general response](https://openreview.net/forum?id=pzUhfQ74c5&noteId=FNXkWfpKC5).
>
> [1]  Prediction-Powered Inference. Anastasios N. Angelopoulos, Stephen Bates, Clara Fannjiang, Michael I. Jordan, Tijana Zrnic. 2023.
>
> [2] Conformal prediction under ambiguous ground truth. David Stutz, Abhijit Guha Roy, Tatiana Matejovicova, Patricia Strachan, Ali Taylan Cemgil, Arnaud Doucet. 2023.
>
> [3] Evaluating content selection in summarization: The pyramid method. Ani Nenkova, Rebecca J Passonneau. 2004.

---

### Official Review · Reviewer_7gqi · 2023-10-31

**Soundness:** 3 good
**Presentation:** 4 excellent
**Contribution:** 3 good
**Rating:** 8
**Confidence:** 3

**Summary:**

This paper studies inference of responses generated by large language models (LLMs),
leveraging ideas of conformal prediction. The specific aim is to construct a set of
responses for a given prompt $x$, such that at least one response in the set is
``acceptable'' (compared with an expert response that we do not get to observe).

The main technical component is from [1], where the Learn then test (LTT) method is
tailored for this specific setting. The authors evaluate the proposed method on the task
of radiology report generation, news summarization, and open-domain question answering.

[1] Angelopoulos, Anastasios N., et al. "Learn then test: Calibrating predictive algorithms to achieve risk control." arXiv preprint arXiv:2110.01052 (2021).

**Strengths:**

1. The paper is very written: it contains sufficient details and is easy to follow.
2. This work provides an interesting application of the LTT framework. From my understanding, the major
contribution includes formulating the task as a risk-controlling problem, specifying the tuning parameters,
and identifying the optimization problem that fits practical goals.
3. The paper also provides abundant numerical evidence supporting the validity and efficiency of the proposed method.

**Weaknesses:**

I do not see obvious weaknesses of this work, but a few questions regarding some details (they are in the question section).

**Questions:**

1. I wonder how the computational time (or relative excess samples) changes with regard to the choice of $k_\max$?
2. In the proposed method, $k_\max$ is fixed. As pointed out by Remark 4.3, the choice of $k_\max$ does not affect
the coverage guarantee, but could result in uninformative results. I wonder if it could be possible to treat $k_\max$ as a tuning
parameter as well, e.g., $\lambda_4 = k_\max$, and we can then optimize over all ``valid'' k_\max.
3. For the individual component pruning, I also wonder if the $\gamma$ can be tuned jointly with the $\lambda$'s to avoid the
union bound?

---

> ### Author Response · Authors · 2023-11-21
>
> We thank the reviewer for their careful review and thoughtful comments. We provide answers to specific questions and remarks (quoted) below.
>
> > I wonder how the computational time (or relative excess samples) changes with regard to the choice of $k_\mathrm{max}$?
>
> Increasing $k_\mathrm{max}$ will decrease the minimum achievable risk, but also increase the computational time (in the worst case). However, even with a high $k_\mathrm{max}$, one can still select a $\hat{\lambda}$ that has similar performance to a different $\lambda$ derived using a lower $k_\mathrm{max}$ (e.g., by placing a higher weight on fewer excess samples in Eq. 7).
>
> One practical advantage of setting $k_\mathrm{max}$ to a lower value is that calibration can be faster. For example, in our implementation we first sample $n \times k_\mathrm{max}$ samples in parallel for the calibration set, and then calibrate $\lambda$ offline.
>
> > I wonder if it could be possible to treat $k_\mathrm{max}$ as a tuning parameter as well
>
> Great suggestion! While it may still be beneficial to explicitly specify a finite upper bound (e.g., select $k_\mathrm{max} \in \mathbb{N} : k_\mathrm{max} \leq K$ for a finite upper bound $K$), it is certainly possible to also tune $k_\mathrm{max}$. In fact, the RAPS method of [1] does something very similar to this idea (where k is a hyper-parameter for improving set quality, which in this case we would also tune during calibration instead of pre-specifying). It would be interesting to explore this in future work.
>
> [1] Uncertainty Sets for Image Classifiers using Conformal Prediction. Anastasios N. Angelopoulos, Stephen Bates, Jitendra Malik, Michael I. Jordan. 2020.
>
> > I also wonder if the $\gamma$ can be tuned jointly with the $\lambda$'s to avoid the union bound?
>
> Yes, this is also another good suggestion! In general, we can use LTT (and more specifically Pareto Testing by Laufer et. al.) to control multiple risks simultaneously. In this case, those risks would be the coverage risk and the false positive risk. $\gamma$ and $\lambda$ can then simply be combined into one configuration, as suggested. That said, as also mentioned in our response to reviewer Sb79, decoupling set construction from component selection makes experimentation and deployment easier (for example, we don't need to redo calibration of components when picking a different $\epsilon$). We found this to be a major practical benefit.

---

### Official Review · Reviewer_zQVz · 2023-11-01

**Soundness:** 2 fair
**Presentation:** 2 fair
**Contribution:** 2 fair
**Rating:** 3
**Confidence:** 2

**Summary:**

This paper proposes a different kind of sampling for LMs. The idea is to extend ideas from Conformal Prediction to produce solution sets that contain diverse outputs. Generation follows a three step process involving sampling, accepting (if the answer is sufficiently high probability), and then possibly resampling if there aren't enough diverse answers in the set yet.

Evaluation is done on a few tasks and LMs -- chest Xrays with a GPT2 small LM + ViT, news summarization on CNN/dailymail with T5-XL, and Open-domain QA on triviaQA sampled from LLaMA-13B. To the best of this reviewer's understanding the evaluation is done assuming access to the ground truth set of references, and so early stopping is done if the prediction set contains the answer already.

**Strengths:**

To this reviewer, the application of conformal prediction to LMs could be interesting. Sampling in a way that preserves diversity and quality; and that returns a variety of different generations at the sequence level, could be a solid contribution. There's a lot of math here (mostly beyond the understanding of this reviewer, who is not an expert on conformal prediction in particular) but that could be helpful to people working in this space, for how to extend these ideas to LMs where the solution space is vast.

**Weaknesses:**

To this reviewer, it's a bit unclear what the benefit of this approach is (ie what problem it's solving). There's a worked example but it's a bit hard to follow as I'm neither an expert in conformal prediction nor cardiology. I guess my confusion comes down to this part:

"Like conformal prediction, our method offers a rigorous coverage guarantee by constructing prediction sets that, in our case, provably contain at least one acceptable response with high probability. Unlike conformal prediction, however, we do not enumerate the entire output space (which is impossible). Instead, we derive a calibrated stopping rule for sampling different outputs from the LM that get added to a growing output set of candidates, until we are confident that the output set is sufficient"

This seems reasonable to this reviewer, but it also seems like from the experiments, the stopping rule involves having access to the test set answers, which seems not realistic. It's not quite clear what the objective is (minimize loss, minimize excess numbers of samples).

I think this paper would also benefit from some stronger yet simpler baselines that don't follow conformal prediction. E.g. sampling N times and checking whether any of the samples is in the ground truth.

I think this paper would benefit from comparison with other types of ways of controlling diversity, e.g. Stochastic Beam Search https://arxiv.org/abs/1903.06059 or NeuroLogic sampling https://aclanthology.org/2022.naacl-main.57/ . A worry I have with this approach (that I asked in the 'questions' field) is that the procedure in Sec 4 might never terminate because the LM might keep returning the same kind of answer over an over again (or at least a sufficiently strong LM might); maybe lexically constrained decoding would fix this.

Finally there's the evergreen concern that this approach might become less important at scale; comparing e.g. GPT2small vs GPT2-XL could help here.

**Questions:**

for the 3-step procedure described in section 4, is it possible this will never terminate? To this reviewer, it seems a little bit sketchy to use ROUGE or BLEU to score diversity.

---

> ### Author Response · Authors · 2023-11-21
>
> We thank the reviewer for their careful review and thoughtful comments.
>
> We want to first clarify one key misconception in this review: **we do not use test set answers as part of our stopping rule.**
>
> The stopping rule is only _calibrated_ using labeled examples, so that it provably generalizes to achieve risk control over future samples with high probability (again, without access to test labels). At test time we only use test labels to evaluate how well our predictions perform. Indeed, the "oracle" with 100% coverage reduces to sampling N times with knowledge of the ground truth set, and picking the first answer that is part of it. Of course this is not realistic, and only serves as a point of comparison (i.e., we measure how many "excess" samples our methods take as compared to this oracle).
>
> We now provide answers to specific questions and remarks (quoted) below.
>
> > It's not quite clear what the objective is (minimize loss, minimize excess numbers of samples).
>
> $\hat{\lambda}$ must provide valid risk control for $C_\lambda$ (so $\hat{\lambda} \in \Lambda_{\mathrm{valid}}$), but should also be a configuration that is useful empirically. Per our validity constraint, the upper bound of our loss is fixed. We are then left with a multi-objective problem where we want to simultaneously minimize set size (number of responses), and the wasted computational effort it takes to generate the set (excess number of samples). We reduced this to a single weighted objective (Eq. 7), with empirically chosen weights. As also indicated to reviewer Sb79, any choice of $\lambda \in \Lambda_\mathrm{valid}$ can also be used without sacrificing our risk control guarantees.
>
> > For the 3-step procedure described in section 4, is it possible this will never terminate?
>
> This is a good question. Simply put, for finite $k_{\mathrm{max}}$, this algorithm will obviously always terminate in finite samples. If $k_{\mathrm{max}}$ is not finite (no sampling upper bound is specified), then with additional assumptions on the underlying LLM and stopping rule (e.g., that the LLM places non-zero probability mass on at least one acceptable answer, which is true for most softmax-based models), we may also show that the algorithm almost surely terminates. However, setting $k_{\mathrm{max}} < \infty$ is more straightforward, and practical.
>
> > Finally there's the evergreen concern that this approach might become less important at scale.
>
> We agree with the reviewer that risks on an absolute level will tend to become smaller at larger model scales.  That said, performance will always be a combination of the base model being used, the hardness of the task at hand, and the desired controlled risk level. This is why it is an important feature that our algorithm is general in nature, and can apply to any model that follows the specified API (i.e., supports sampling). This supports  independent research in improving the underlying LM, in addition to the scoring functions used, while keeping the pipeline the same. In fact, it is great if the underlying LM is better: this will increase the range of achievable risks, and will decrease both the average size of the prediction set, and the number of samples required to generate to derive it.

---

> ### Author Response · Authors · 2023-11-22
> **Discussion**
>
> As the discussion deadline is approaching, we would like to know if we have addressed your comments, or if there is anything else we can help to clarify. Thank you again for taking the time to review our work.

---

> > ### Comment · Reviewer_zQVz · 2023-11-23
> > **thanks! keeping my score**
> >
> > thanks for the detailed response! I'd like to keep my score - I feel like to this reviewer, it's still not quite clear what problem this is solving or how much the approach proposed would outperform simpler baselines.

---

### Official Review · Reviewer_Sb79 · 2023-11-10

**Soundness:** 3 good
**Presentation:** 2 fair
**Contribution:** 3 good
**Rating:** 6
**Confidence:** 4

**Summary:**

This paper extends the conformal prediction framework to language models (LMs). The aim is to construct prediction sets of LM responses such that at least one is "acceptable" with a desired probability.

This goal is achieved by sampling responses repeatedly from the LM and adding them to the prediction set until a stopping rule occurs: either (i) an upper bound $k_{\max}$ on the number of samples drawn from the LM, or (ii) a lower bound $\lambda_{3}$ on the prediction set confidence score, quantifying how good it is. Additionally, samples are removed from the prediction set (to reduce noise) based on a rejection rule: either (i) a lower bound $\lambda_{2}$ on the individual response's quality, or (ii) an upper bound $\lambda_{1}$ on the response's similarity with others in the set. $\lambda = (\lambda_{1}, \lambda_{2}, \lambda_{3})$ is a configuration that is picked via the Learn Then Test [Angelopoulos et al. 2021a] framework; $k_{\max}$ is fixed by the user. Note that not all desired errors are achievable. The paper also proposes an extension to identify individual components of the responses (for instance, sentences from paragraphs) that are each independently "acceptable" with desired probability.

The paper goes on to provide empirical results to corroborate its methodology. Experiments on open-domain question-answering, text summarization, and radiology report generation empirically verify the proposed algorithm and its statistical guarantees.

**Strengths:**

1. This paper tackles an important research area of providing statistical guarantees on the output of language models.
2. These statistical guarantees are provided by leveraging the conformal prediction framework and extending it to language models, which is a generative task.
3. The proposed algorithm is easy to interpret: sample responses from the language model as long as the rejection and the stopping rules are not triggered.
4. The theoretical results are corroborated experimentally on 3 tasks.

**Weaknesses:**

[Details in the Questions section]

1. The proposed algorithm is easy to interpret. However, the paper does not explain all design choices.
2. The experimental results support Theorem 4.2. However, Proposition 4.4 is not experimentally validated.
3. Theorem 4.2 guarantees at least one "acceptable" language model response in the prediction set but does not say much about the individual responses. While the paper considers components of the responses, the response as a whole is not.

**Questions:**

1. Design choices:
    1. Why is $\hat{\lambda}$ defined as in Eq. 7? Similarly, why is it desirable for $C^{\text{inner}}_{\gamma}$ to be large ($\hat{\gamma}$ in Eq. 10)? Explicitly explaining these would help the reader. Are there alternative definitions that might be advantageous?
    2. What is the reason for using a different definition of $C^{\text{inner}}_{\gamma}$ during calibration?
    3. The Pareto Testing procedure is used in the proposed algorithm for efficiency. However, the appendix only provides a high-level overview. Can the authors include more details and explain how this procedure helps efficiency?

2. Experiments:
    1. The achievable risk range for MIMIC-CXR is very high. How can one reduce this for the practicality of the proposed method?
    2. How is the AUC for the set loss plots greater than 0.5? The diagonal should have an AUC of 0.5, and since all the curves are under this, their respective AUCs should be lower than 0.5.
    3. The First-$k$ scoring function does not utilize rejection. As a result, the likelihood-based approaches have better prediction efficiency. Did the authors try First-$k$ with the rejection scheme for a more apples-to-apples comparison?
    4. What about the set loss in the individual component case? Experimental results on that would corroborate the theoretical result in Proposition 4.4.
    5. What is $k_{\max}$ set to for the experiments?
    6. The data splits to obtain the train, calibration, and test sets aren't well explained in the paper (appendix included). For example, the authors use a dev set for the conformal prediction experiments, but in what way? Can the data splits be explained more succinctly for the datasets used?

3. "While Eq. 1 stipulates the existence of at least one "acceptable" generation in $C_{\lambda} (X_{\text{test}})$, it does not tell us much about individual responses, $y \in C_{\lambda} (X_{\text{test}})$" (Section 1). Is there any remedy for this? This paper discusses individual components (for example, sentences) but not the whole LM responses.

Clarifications:

1. It would be helpful for the reader to highlight why admission functions $A_{i}$ are used rather than labels $Y_{i}$ in Section 1 (where the setup is explained).
2. Shouldn't the function $A^{c}$ be a mapping $\mathcal{Y} \mapsto \\{0, 1\\}$ instead of $2^{\mathcal{Y}} \mapsto \\{0, 1\\}$ (Section 4.4) as it acts on individual components?
3. $\mathcal{Y}$ and $\mathcal{V}^{*}$ are used interchangeably. Making it consistent would help the reader.
4. It would be helpful for the reader to add more descriptive captions for the tables in Appendix H. For example, the authors can highlight what the findings are not just in the text but in the table captions as well.
5. Should Eq. 20 be replaced with Eq. 7?
6. Missing definitions:
    1. $\mathcal{Y}$ is not defined in Section 1 but is used right before Eq. 1.
    2. $L_{\text{test}} (\lambda)$ is not defined but is used right before Eq. 3.
    3. $F_{X} (u), F_{Y} (y)$ are not defined in Appendix D.1.
7. Missing superscripts:
    1. The superscript $c$ in $A^{c}_{\text{test}}$ is missing in the text before and after Eq. 9.
    2. The superscript $c$ in $\bar{L}^{c} (\gamma)$ is missing in the text after Eq. 22.

---

> ### Author Response · Authors · 2023-11-21
>
> We thank the reviewer for their careful review and thoughtful comments. We provide answers to specific questions and remarks (quoted) below.
>
> > The paper does not explain all design choices.
>
> Thank you for highlighting where our design choices were not well explained. We have improved the clarity of our paper with respect to these questions (as detailed below).
>
> > Proposition 4.4 is not experimentally validated.
>
> We initially left this out for sake of space; we have now added a plot demonstrating empirical validity---see Figure G.1.
>
> > Theorem 4.2 guarantees at least one "acceptable" language model response in the prediction set but does not say much about the individual responses.
>
> This is consistent with the style of guarantees that standard conformal inference / confidence intervals give. It is also good to keep in mind that relative rankings/confidence scores of responses in the prediction set (e.g., by using our "\mathcal{Q}" individual response quality function) can still be computed, though we just don't have formal _guarantees_ on their individual correctness (again, the statements in Theorem 4.2 only apply to the set as a whole). Notably (and different from standard conformal prediction due to the structure of the problem we are dealing with), our component-level prediction does provide _guarantees_ about sub-predictions relevant to individual responses, that hold simultaneously across all responses in the set.
>
> > Why is $\hat{\lambda}$ defined as in Eq. 7? Similarly, why is it desirable for $C_{\gamma}^{\mathrm{inner}}$ to be large?
>
> $\hat{\lambda}$ is selected to provide valid risk control for $C_\lambda$ (so $\hat{\lambda} \in \Lambda_{\mathrm{valid}}$), while also being the configuration that is most useful empirically. That means, we want $C_\lambda$ to be precise (contain as few items as is necessary), and we want $C_\lambda$ to be quick to evaluate (not require sampling many responses before the stopping criterion is met). The combined objective we specify for $\hat{\lambda}$ therefore reflects a tradeoff between these two qualities. **We have clarified this in the paper.**
>
> Meanwhile, $C_\gamma^{\mathrm{inner}}$ identifies the set of subcomponents that our model is very confident in, while controlling for false positives. Ideally, this would contain _all_ correct subcomponents (i.e., we have 100% recall of correct subcomponents). Simply making $C_\gamma^{\mathrm{inner}}$ as large as possible, while subject to our constraint, helps achieve maximum recall. **We have also clarified this in the paper.**
>
> > What is the reason for using a different definition of $C_\gamma^{\mathrm{inner}}$ during calibration?
>
> This decouples set construction from component selection, and makes experimentation/deployment easier (i.e., we don't need to redo component calibration for every different $epsilon$). In practice, we can do either one. **We have clarified this in the paper.**
>
> > The Pareto Testing procedure is used in the proposed algorithm for efficiency. However, the appendix only provides a high-level overview.
>
> We have added a detailed description of Pareto testing to Appendix E.
>
> > The achievable risk range for MIMIC-CXR is very high. How can one reduce this for the practicality of the proposed method?
>
> This can be reduced by either increasing $k_{\mathrm{max}}$ (which allows for more sampling effort), or by improving the underlying language model. Our calibration method applies to any base model or $k_{\mathrm{max}}$.
>
> > How is the AUC for the set loss plots greater than 0.5?
>
> Thanks for raising this question; we realize that we did not fully explain how we compute the AUC in the paper. Specifically, we compute a _normalized_ AUC as $\frac{1}{b - a}\int_{a}^{b} f(x) dx$, where the limits $a$ and $b$ are set to the minimum achievable epsilon and to the trivial epsilon (the risk of the top-1 prediction), respectively. This normalized AUC can be greater than 0.5. **We have clarified this, and have included the values for the diagonal in the plots.**
>
> > The First-$k$ scoring function does not utilize rejection.
>
> Thanks for the suggestion—we added a variant with rejection. Please refer to our [general response](https://openreview.net/forum?id=pzUhfQ74c5&noteId=FNXkWfpKC5).
>
> > What is $k_{\mathrm{max}}$ set to in the experiments?
>
> It is set to 20 (Appendix C). We have moved this to the main text.
>
> > The data splits to obtain the train, calibration, and test sets aren't well explained in the paper.
>
> We have clarified these details in Appendix F.4.
>
> Additional:
>
> - We have switched to using $\mathcal{Y}$ throughout the paper now for simplicity. We still discuss the structure of $\mathcal{Y}$ in Section 4.1.
> - Thank you for catching the typo for $A^c$! We have corrected it.
> - We have added more descriptive captions to the tables in Appendix H.
> - Thank you for catching the typo in Eq. 20! We have updated it to be consistent with Eq. 7.
> - We have fixed the missing definitions and subscripts.

---

> > ### Author Response · Authors · 2023-11-22
> > **Discussion**
> >
> > As the discussion deadline is approaching, we would like to know if we have addressed your comments, or if there is anything else we can help to clarify. Thank you again for your helpful comments, and for taking the time to review our work.

---

### Author Response · Authors · 2023-11-21
**Main Author Response**

Thank you to all the reviewers for taking the time to read and comment on our work. We were pleased to receive several good suggestions, and have taken this feedback into account. Please see our revised manuscript.  Major revisions to the text are highlighted in red.

Most notably, we have also added experimental results for using a First-K stopping rule together with rejection, as suggested by reviewers. To clarify, our original First-K baseline is meant as a simple case with a single dimensional parameter ($k$). $\hat{\lambda} = \hat{\lambda_1} \in \mathbb{N}$ is therefore very simple to select and reason about. Now, we have also paired this (deterministic) stopping rule with the ability to reject duplicates and low-quality samples using the same scoring tools as the Max and Sum methods. This now has three parameters ($\lambda_1$ for the total number of samples to take, and $\lambda_2$ and $\lambda_3$ for the rejection rules) and requires Pareto Testing for efficient calibration, but isolates the effects of the stopping rule (dynamic stopping based on Max/Sum vs. static stopping after taking the first K samples). As expected, rejection helps reduce the final set size, but still lags in computational efficiency (the number of excess samples is high).

We have responded to the remainder of the comments raised by reviewers individually. Please let us know if any questions or concerns remain. We look forward to additional discussion.

---

### Meta-Review · Area_Chair_31CT · 2023-12-06

**Metareview:**

This paper proposes a conformal approach to generating a set of responses with language models (LMs). The main contribution is an algorithm that can control a specific set-valued risk (the expected indicator loss that it contains at least one "acceptable" response according to an "admission function"), while in the mean time maximizing some other quantities of interest such as quality, diversity, and confidence (each with a customizable metric). The algorithm adapts the Learn-Then-Test framework. The algorithm is empirically tested on three language generation tasks, and also extended to handle a component selection problem.

Overall, the paper appears to be well executed, and the proposed algorithm is (at least conceptually) broaderly useful---basically, it gives an acceptance/rejection/stopping rule for LM generation that can control a certain set-valued risk, and in addition maximize certain additional criteria. Therefore, I recommend acceptance.

As suggested by the reviewers, there are some concerns about too many pieces/design choices, and confusions about the objective of the approach (may want to highlight the risk control as the main objective and the three quality criteria as secondary). I also feel like the dense notation in the first 7.5 pages may limit its digestibility to a broader audience, should such broader adoption be a goal of the paper. I encourage the authors to revise the presentation to alleviate these issues in the final version.

**Justification For Why Not Higher Score:**

The calibration procedure may require strong problem assumptions, in particular a calibration set where the admission functions are known.

**Justification For Why Not Lower Score:**

The paper gives a acceptance/rejection/sampling rule for LM sampling with risk control guarantees, which could be broaderly useful.

---

### Decision · Program_Chairs · 2024-01-16

Accept (poster)